# Evidence for the predictability of changes in the stratospheric aerosol size following volcanic eruptions of diverse magnitudes using space-based instruments

Larry W. Thomason[1], Mahesh Kovilakam[2], Anja Schmidt[3,4], Christian von Savigny[5], Travis Knepp[1], Landon Rieger[6]

[1]NASA Langley Research Center, Hampton, Virginia 23681 USA
[2]SSAI, Hampton, Virginia, USA
[3]Department of Chemistry, Cambridge University, Cambridge, UK
[4]Department of Geology, Cambridge University, Cambridge, UK
[5]Institute of Physics, University of Greifswald, Greifswald, Germany
[6]University of Saskatchewan, Saskatoon, Saskatchewan, Canada

*Correspondence to*: Larry W. Thomason (l.w.thomason@nasa.gov)

**Abstract.** An analysis of multiwavelength stratospheric aerosol extinction coefficient data from the Stratospheric Aerosol and Gas Experiment II and III/ISS instruments is used to demonstrate a coherent relationship between the perturbation in extinction coefficient in an eruption's main aerosol layer and the wavelength dependence of that perturbation. This relationship spans multiple orders of magnitude in aerosol extinction coefficient of the stratospheric impact of

volcanic events. The relationship is measurement-based and does not rely on assumptions about the aerosol size distribution. We note limitations on this analysis including that the presence of significant amounts of ash in the main sulfuric acid aerosol layer and other factors may significantly modulate these results. Despite this limitation, these findings suggest an avenue for improving aerosol extinction coefficient measurements from single channel observations such as

the Optical Spectrograph and Infrared Imager System as they rely on a prior assumptions about particle size. They may also represent a distinct avenue for the comparison of observations with interactive aerosol models used in Global Climate Models and Earth System Model.

## 1 Introduction

Volcanic eruptions represent the primary source of variation in stratospheric aerosol levels

(Thomason et al., 1997b;Solomon et al., 2011;Schmidt et al., 2018;Robock, 2000).  The optical

signature of volcanically-derived aerosol is generally dominated by sulfuric acid droplets but this

can be enhanced by the presence of ash either mixed with the sulfuric acid droplets or as distinct

layers (Winker and Osborn, 1992;Vernier et al., 2016). Sulfuric acid aerosol are known for its

ability to significantly modulate climate (Schmidt and Robock, 2015) primarily by scattering

incoming solar radiation to space and even relatively small volcanic events have been noted to

affect global temperature trends (Santer et al., 2014).  In addition, since sulfuric acid aerosol

particles absorb upwelling infrared radiation, the presence of a volcanic aerosol layer can change

the thermal structure of the stratosphere (Labitzke, 1994) and the troposphere and modulate

stratospheric circulation as well as transport across the tropopause (Pitari et al., 2016).

Significant effort has been expended toward measuring stratospheric aerosol by a variety of

instruments (Kremser et al., 2016) and an extensive data collection of observations are now

available. Some Global Climate Models (GCMs) and Earth System Models (ESMs) use these

measurements or parameters derived from them directly (Mann, 2015) while others, that use

interactive aerosol model schemes (Mills et al., 2016) and similar tools (Toohey et al., 2016),

assess how well their tools replicate observations and thus infer the reliability of the models

assessment of the climate impact of volcanic eruptions (Timmreck et al., 2016).

The initial impetus for this study was to develop tools to understand how reliably the long-term

variability of stratospheric aerosol can be characterized given the limited data sets available.

Thus, one aim of this work was to understand how small-to-moderate volcanic events manifest

themselves in SAGE II/III observations with the goal of inferring the uncertainty in single

wavelength space-based data sets that use a fixed aerosol size distribution as a part of their

retrieval algorithm such as OSIRIS (2002-present) and the Cloud-Aerosol LIdar with Orthogonal Polarization (CALIOP; 2006-present)  (Rieger et al., 2019;Kar et al., 2019). The current OSIRIS algorithm is dependent on a priori assumptions about the aerosol size distribution and thus a fixed spectral dependence for aerosol extinction coefficient. As we show below, there are substantial changes in the spectral dependence of aerosol extinction coefficient following these eruptions which the current OSIRIS algorithm does not capture. A longer-term goal is to infer how well the wavelength dependence can be estimated for these single wavelength measurements. These factors are relevant, not only to understanding the limitations in single channel data sets, but also to multi-instrument data sets that are reliant on them such as the Global Space-based Stratospheric Aerosol Climatology (GloSSAC) (Kovilakam et al., 2020). For this study, we make use of observations made by the Stratospheric Aerosol and Gas Experiment (SAGE) II (1984-2005) and III/ISS (2017-present) which span a broad range of volcanic perturbations of the stratosphere. We demonstrate that, for the most part, the changes in aerosol extinction coefficient and apparent aerosol particle size, where we use the spectral dependence of aerosol extinction coefficient as a proxy for size, are well correlated across nearly 2 orders of magnitude in extinction coefficient change.  This relationship is a directly measurable characteristic of the changes in aerosol size distribution following an eruption without assumptions regarding the functional form for the aerosol size distribution (e.g., log-normal). Since comparisons of interactive aerosol model scheme calculations and measurements of stratospheric aerosol form the basis of assessing the performance of these aerosol microphysics modules, the observed relationship provides a potentially unique, measurement-focused means for assessing interactive aerosol models for volcanic eruptions of different magnitudes.

**2 Data and Methods**

Space-based measurements of stratospheric aerosol have been made on a nearly global basis

since the Stratospheric Aerosol and Gas Experiment (SAGE) aboard the Applications Explorer

Mission 2 platform operated from 1979 through 1981 (Chu and McCormick, 1979). The SAGE

II mission (https//doi.org/ 10.5067/ERBS/SAGEII/SOLAR_BINARY_L2-V7.0) spanned the

recovery of stratospheric aerosol levels from two large magnitude volcanic eruptions the eruption

of El Chichón in 1982 and the 1991 eruption of Mt. Pinatubo (Thomason et al., 2018). Here we

define large-magnitude eruptions as those with a Volcanic Explosivity Index (VEI; Newhall and

Self, 1982) of 6 or more, and small-to-moderate-magnitude eruptions as those with a VEI of 3, 4,

or 5 whereby we only consider those eruptions that had a measurable impact on the stratospheric

aerosol load in the period 1979 to 2019 (see Table 1).The Mt. Pinatubo eruption was the largest

stratospheric event since at least Krakatau in 1883 (Stothers, 1996). In the SAGE II record, the

Mt. Pinatubo event remains clearly detectable until the late 1990s and thus it has an impact on

nearly half of the 21-year dataset. In the seven years of SAGE II observations prior to Mt.

Pinatubo, stratospheric aerosol levels consistently decrease following the 1982 El Chichón

eruption (Thomason et al., 1997a). As a result, nearly 75% of the SAGE II record is dominated

by the recovery from two large magnitude volcanic events.  This can be clearly seen in Figure 1

where the long-term variation of stratospheric aerosol optical depth from the Global Space-based

Stratospheric Aerosol Climatology (GloSSAC), a global multi-instrument climatology of aerosol

optical properties,  is shown for 1979 through 2018 (Kovilakam et al., 2020). As a result, due to

the timing of the SAGE II mission, much of what is inferred as the 'normal' properties of

stratospheric aerosol inferred from SAGE II observations is skewed toward these large events

rather than a handful of small-to-moderate events that occur throughout the period of interest.

As shown in Figure 1, starting with the January 2005 eruption of Manam, which is near the end of the SAGE II record (October 1984 through August 2005), there are regular injections of aerosol and its precursors following volcanic eruptions. While none of these events approach the magnitude of Mt. Pinatubo or El Chichón, they were able to subtly modulate climate and are of general scientific interest (Solomon et al., 2011;Ridley et al., 2014;Schmidt et al., 2018)  From the end of the SAGE II mission in August 2005 until the start of the SAGE III/ISS mission in June 2017, space-based missions consist of measurements used in GloSSAC from instruments such as  OSIRIS and CALIOP (Rieger et al., 2019;Kar et al., 2019) and data from other instruments including SCIAMACHY (von Savigny, 2015), MIPAS (Griessbach et al., 2016), OMPS LP (Loughman et al., 2018) and GOMOS (Bingen et al., 2017).  Since the start of the on-going SAGE III/ISS mission in June 2017 (https//doi.org/10.5067/ISS/SAGEIII/SOLAR_HDF4_L2-V5.1), several additional small-to-moderate volcanic events have been observed including two eruptions by Ambae in April and July 2018 (Kloss et al., 2020b), Raikoke (June 2019) (Muser et al., 2020), and Ulawun (June/August 2019) .  In addition, there are at least two pyrocumulus (also known as flammagenitus) events, particularly the Canadian forest fire event of August 2017 (Kloss et al., 2019;Bourassa et al., 2019) and the Australian bush fires of December 2019 and January 2020 (Khaykin et al., 2020). The non-volcanic events are interesting but not the focus of this paper.

After 2005, the frequency of small volcanic and smoke events is substantially higher than observed during the SAGE II mission and there is a significant qualitative difference in the stratospheric aerosol variability in between the two periods. After the end of the SAGE II mission in 2005 and until the start of the SAGE III mission, the long-term stratospheric record is

less robust partly due to the limited global multiwavelength measurements of aerosol extinction

coefficient.

It should be clear from the outset that the solar occultation measurement strategy is, in general, not conducive to process studies and understanding the distribution of aerosol following highly localized events like volcanic eruptions. Following these sorts of events, we observe that SAGE observations have a high zonal variance in the data compared to more benign periods where the

zonal variance is often not much larger than the measurement uncertainty particularly in the tropics (Thomason et al., 2010). The events we discuss below are not sampled in a temporally uniform way and the time between an eruption and the first SAGE II observations at the relevant latitudes varies from a few days to more than a month.  This is an outcome of the sparse spatial sampling characteristic of solar occultation with latitudinal coverage dictation by orbital and

seasonal considerations and a given latitude is measured at best once or twice per month. In addition, with 15 profiles per day with 24 degrees of longitude spacing, the sampling is sparse in longitude even when latitudes of interest are available.  Furthermore, aerosol properties in a single profile at a single altitude are the average of multiple samples along different line-of-sight paths through the atmosphere such that the spatial extent of a measurement at an altitude extends

over hundreds if not thousands of square kilometers (Thomason et al., 2003).  This large measurement volume increases the possibility that only part of a SAGE II observation's measurement volume that will actually consist of a mix of volcanically-derived material and unperturbed stratosphere. As a result, the interpretation of an extinction measurement pair must be interpreted similar to the way that SAGE observations of water clouds are better interpreted as

aerosol/cloud mixed extinction coefficient values rather than purely 'cloud' extinction coefficient

(Thomason and Vernier, 2013). With these limitations, the ability to characterize the attributes of the early plume is limited.

The SAGE instruments use solar occultation to measure aerosol extinction coefficient at multiple wavelengths from the UV to the near infrared. These measurements are of high accuracy and precision across a broad range of extinction levels and have a vertical resolution of ~1 km and are reported in 0.5 km increments from 0.5 to 40.0 km (Damadeo et al., 2013). The multi-wavelength aerosol extinction coefficient measurements provide limited information regarding the details of the aerosol size distribution of the aerosol (Thomason et al., 2008;Von Savigny and Hofmann, 2020) though many efforts at deriving the aerosol size distribution have been proposed (Yue and Deepak, 1983;Wang et al., 1996;Bingen et al., 2004;Malinina et al., 2018;Bauman et al., 2003;Anderson et al., 2000). The primary measure of particle size for SAGE II comes from the ratio of the aerosol extinction coefficient measurements at 525 and 1020 nm. Figure 2a shows the Mie aerosol extinction coefficient as a function of particle radius at 525 and 1020 nm for sulfuric acid aerosol at stratospheric temperatures (based on Bohren and Huffman (1998)) and their ratio is shown in Figure 2b . While incorporating a realistic size distribution would complicate the picture, the ratio relationship shows approximately how the inferred aerosol size changes with extinction coefficient ratio. Over the lifetime of the SAGE II mission, in the stratospheric aerosol layer, this ratio varies from around 5 (~0.2 μm) to values around 1 where the ability to discriminate aerosol is reduced to noting that the particles are 'large' with extinction dominated by aerosol larger than ~0.5 μm. As shown in Figure 3, the mean GloSSAC v2.0 525-nm stratospheric aerosol optical depth between 20°S and 20°N, whose construction is discussed in detail in Kovilakam et al. (2020), increased between June and July 1991 by a factor of about 40. At the same time, the 525 to 1020-nm optical depth ratio changed from around 3.3

to a ratio of about 1.2. With low volcanic activity in this period, the relaxation of stratospheric

aerosol loading toward background levels remains obvious in the tropics into the late 1990s.

The Mt. Pinatubo event can lead to the perception that the 'normal' process is that volcanic input

into the stratosphere generally increases aerosol extinction coefficient and decreases aerosol

extinction coefficient ratio (suggesting an increase in the size of particles that dominate aerosol

extinction). However, we will demonstrate below that the impact of volcanic events on

stratospheric aerosol extinction coefficient ratio is strongly modulated by the magnitude of the

eruption and, to a lesser extent, the stratospheric aerosol loading prior to the eruption. We will

also show that the data suggests that sulfur rich but relatively ash-poor eruptions show a

consistent, predictable behavior that lends itself as a test for interactive aerosol schemes used in

global climate models. We also observe that the presence of large aerosol, probably ash,

following a few eruptions significantly modulate these results.

**3 Results**

Herein, we examine the impact of 11 eruptions by 9 volcanoes (see Table 2) that affected the

stratosphere for which there are SAGE II or SAGE III/ISS measurements available. These begin

with the November 1985 eruption of Nevado del Ruiz (Colombia) and continue to the second

eruption of Ulawun (Papua New Guinea) in August 2019. Two volcanoes have two eruptions in

this record: Ambae in April and July 2017 and Ulawun in June and August 2019. Due to the

nature of SAGE III sampling the Ulawun events cannot be distinguished well and are treated as a

single event. Overall, the eruptions increase aerosol extinction coefficient between $10^{-4}$ and $10^{-2}$

km$^{-1}$ relative to pre-eruption levels with a similar two order of magnitude relative increase

compared to the levels observed prior to the eruptions. From observations in the latitude region

near the location of each eruption and extending from just prior to each eruption and continuing

for several months following, we infer the impact of these eruptions by noting the perturbation on the stratospheric aerosol extinction at both 525 and 1020 nm when the extinction coefficient at 1020 nm is a maximum. The ratio of these perturbations provides a rough assessment of the

impact of the eruptions on the size of particles dominating aerosol extinction. We analyze data from SAGE II and SAGE III/ISS in identical ways except for one detail. The current version of SAGE III data (5.1) has a defect in which aerosol extinction at 521 nm is biased low below about 20 km due to an error in the $O_4$ absorption cross section used in processing this version. The $O_4$ error has a subtle, positive impact on the ozone retrieval below 20 km where there is significant

overlap in the spectral regions used to retrieve ozone and where $O_4$ absorbs. The small error in ozone has a larger impact on aerosol where ozone absorbs strongly (521, 602 and 676 nm) but other aerosol measurement wavelengths are unaffected. Therefore, we have replaced the 521 nm data product with an interpolation between 448 and 756 nm that employs a simple Angstrom coefficient scheme. The 448 and 756 nm aerosol extinction coefficient do not manifest the bias

while 602 nm and 676 nm measurements have biases similar to those at 521 nm. The interpolation is possible since the stratospheric aerosol extinction coefficient is always observed to be smoothly varying with wavelength and approximately linear in log-log space. The presence of the 521 nm bias is inferred using this methodology and this approach was used in the validation paper for SAGE III/Meteor 3M aerosol data (Thomason et al., 2010). The differences

between the inferred 521 nm extinction coefficients and the reported values in the lower stratosphere (tropopause to 20 km) average about 6% and are usually less than 10%. Above 20 km the differences are usually on the order of 1 to 2% with the estimate usually less than the observation that is probably a reflection of the limitation of the accuracy of the interpolation and consistent with past uses of the same approach (Thomason et al., 2010). In any case, the same

arguments on the effects of small to moderate volcanic eruptions on aerosol extinction

coefficient as a function of wavelength described below can be made whether 448 or 521 nm

aerosol extinction coefficient is used in the SAGE III analysis.  We interpolate the 521 nm values

solely for comparison purposes with SAGE II data and this process has minimal impact on the

conclusion drawn below.

For each event, we collect all SAGE II/III aerosol extinction coefficient data at 525 and 1020 nm

between 10 and 25 km where the profiles occur within 10 degrees of latitude of the eruption for a

period starting 3 months prior to the eruption through 6 months following it. Depending on the

latitude, as recorded in Table 2, and season, the volume and frequency of observations can vary

significantly. Figure 4a shows all the data for Nevado del Ruiz in this temporal window at the

altitude of the maximum increase in aerosol extinction coefficient, in this case 20.5 km. The

Nevado del Ruiz eruption occurred on 13 November 1985 (Julian day 317) and the immediate

enhancement of aerosol extinction coefficient is clear as aerosol extinction coefficient increases

by about an order of magnitude from about 0.0007 km$^{-1}$ to values approaching 0.01 km$^{-1}$. As

shown in Figure 4b, the aerosol extinction coefficient ratio increases from about 2.2 prior to the

eruption to a broad range of values from 2 to 3.5 immediately following the eruption (~day 380

or January 1986) and in opposite sense of what was observed following the Mt. Pinatubo

eruption as shown in Figure 3. The Nevado del Ruiz extinction ratio becomes much more

consistent in the subsequent samples of this region of the stratosphere and falls from roughly 2.8

to 2.4 at the end of the analysis period (~day 560 or July 1986).  The spread early in extinction

coefficient and in extinction coefficient ratio is primarily due to inhomogeneity in the volcanic

aerosol within the analysis area (Sellitto et al., 2020). This is suggested by Figure 5 in which the

extinction coefficient ratio is plotted versus the extinction coefficient for this data set. Almost

without exception, the enhancement in aerosol extinction coefficient is associated with larger values of extinction coefficient ratio.  The distinction between volcanically perturbed observations and the unperturbed periods prior to the eruption is clearly recognizable. A handful of points show very high aerosol extinction coefficients but extinction coefficient ratios close to and occasionally less than those observed prior to the eruption (<2.3 or so). For these observations, some large particles (possibly ash) are evidently present but, since SAGE-like observations contain little or no information about composition, their composition cannot be inferred unambiguously. In any case, these points are rare and only observed in the first month following the eruption possibly due to the removal of large particles by sedimentation. Generally, we find that the low latitude eruptions like Nevado del Ruiz exhibit zonal variability in aerosol extinction coefficient than mid and high latitude events.  For instance,  SAGE III/ISS observations of the Canadian pyrocumulus event of August 2017 (Bourassa et al., 2019)  varied in extinction coefficient at some latitudes from pre-event extinction of $10^{-4}$ km$^{-1}$ to values that exceeded $10^{-2}$ km$^{-1}$ as late as the end of October 2017.  In this regard, low latitude events are a more straightforward evaluation than high variability, higher latitude events.

Given the geometry of the solar occultation measurements, SAGE II and III sample a latitude band episodically, revisiting a latitude every few weeks to months making observations in a latitude band for 1 to several days. This sampling pattern is clear in Figure 4a and 4b.  We defer to this pattern and average the extinction values at both 525 and 1020 nm into these irregularly spaced and duration temporal bins. We required a minimum of 6 profiles to be available in the temporal bin to be included in further analysis. This eliminates a few periods such as the few points around Julian day 340 and again around Julian day 350 as seen in Figure 4a. Within each bin, we select the maximum values of extinction coefficient at 1020 nm in each profile within a

4-km vertical window (9 observations) extending from 1 km below to 3 km the broadly observed

maximum in the extinction profiles (20.5 km in this case) as we try to capture the behavior of the

most intense part of the volcanic layer including a tendency for the layer to increase in altitude

during the months following the eruption. The 4-km window is primarily a way to find the

altitude (and the associated extinction coefficients) of the volcanic layer in each profile where it

can vary from profile to profile within a temporal bin and over the months following the

eruption. For events in this analysis, there is a 0.5 to 2 km rise in the altitude of peak aerosol

extinction coefficient during the analysis period following the eruption due mostly to dynamical

processes (Vernier et al., 2011). The averaging produces a simplified characterization of the

effects of the eruption as shown in Figure 6. In this figure, we see that the change in aerosol

extinction coefficient and extinction coefficient ratio are well correlated with both reaching a

maximum near Julian day 380 (as sampled by SAGE II). One difference is that while both

parameters begin to relax back toward pre-eruptive levels, the extinction coefficient does so quite

a bit more quickly than the extinction coefficient ratio. Since the scale for the extinction

coefficient ratio does not extend to zero, the difference in the recovery rates is even more

significant. Figure 7 shows the same plots for the remaining nine eruptions.  They can be crudely

sorted into two categories. While all show relatively rapid increases in aerosol extinction

coefficient at 1020 nm with the maximum values occurring with the first or second observation

by SAGE II/III, one category of eruption are similar to the Nevado del Ruiz eruption with rapid

increases in aerosol extinction ratio following the eruption. These tend to be among the smaller

eruptions and include: Cerro Hudson in 1991 (Figure 7c), Manam in 2005 (Figure 7e), Ambae

twice in 2018 (Figure 7f), and Ulawun twice in 2019 (Figure 7g). In the case of the second

Ambae eruption, there is a small increase in the observed aerosol extinction coefficient ratio

following the eruption and it remains large (~4.8) compared to the value prior to the first Ambae

eruption (~3.2). A second category of volcanic events show the opposite behavior with a

decrease in extinction ratio following an event including Kelut in 1990 (Figure 7a), Mt. Pinatubo

in 1991 (Figure 7b), Ruang in 2002 (Figure 7d), and Raikoke in 2018 (Figure 7g).  We will now

discuss some individual events.

Figure 8a shows the before-and-after state of the main aerosol layer for these 10 eruptions where

'before' values are defined as the first data point in the series shown in Figure 7 and the 'after' is

defined where the 1020 nm aerosol extinction coefficient reaches a maximum.  As one could

infer from Figure 7, we see two types of events, those with positive slopes (larger

extinction/larger extinction ratio) and those with negative slopes (larger extinction/smaller

extinction ratio) with some suggestion of a change of slope from strongly positive to negative

with increasing aerosol extinction coefficient perturbation. To isolate this change, we define an

aerosol extinction coefficient perturbation to be

$$\delta k_\lambda = k_\lambda(after) - k_\lambda(before) \qquad (1)$$

which is computed for 1020 and 525 nm where 1020 nm aerosol extinction coefficient is a

maximum. It should be noted that the maximum extinction coefficient at 525 nm does not

necessarily occur at the same altitude or time as the maximum in 1020 nm extinction coefficient.

There is some variability in the timing of  the 'before' data used in this analysis, however, within

these data sets, we observe that aerosol extinction coefficient levels at a given altitude and

latitude slowly vary with time independent of recent volcanic activity due to the recovery from

past volcanic activity and seasonal processes. For the events discussed here, due to the timing of

the events, these changes are very small compared to the volcanic events in our study and, in

terms of the calculation of perturbation values, the exact background level has only a secondary

effect on the calculated values.   As a result, the timing of the 'before' samples does not

materially affect these results. We define an aerosol extinction coefficient perturbation ratio (or

more simply perturbation ratio) as

$$perturbation\, ratio = \delta k_{525}/\delta k_{1020}. \qquad (2)$$

Figure 8b shows the relationship between the perturbation parameters. The perturbation ratio for

8 of these events is well sorted by the magnitude of the extinction coefficient perturbation from

the smallest extinction coefficient perturbation event (Manam) and the largest (Mt. Pinatubo).

Based on Figure 2b, we would expect that the relationship would asymptote to about 1 for large

events near or larger than Mt. Pinatubo, reflecting the presence of very large radius aerosol (>0.4

μm) so some sort of curvature seems reasonable.  It should be noted that SAGE II did not

observe the entirety of the Mt. Pinatubo plume due to its extreme opacity. However, the

observations available uniformly show very high extinction ($>10^{-2}$ km$^{-1}$) and low extinction ratio

(~1) with all observations. So while the detailed location of Mt. Pinatubo data in plots 7 and 8 is

not exact, the general location particularly in Figure 8b is representative of this event. While the

perturbation ratio approach effectively treats the aerosol as an add-on to the 'before' aerosol

extinction, we do not suggest that volcanic aerosol does not interact with the pre-existing aerosol.

Nonetheless, the observed relationship in Figure 8b suggests that the values of the perturbation

pair (extinction coefficient and perturbation ratio) are insensitive to the initial conditions of the

stratospheric aerosol. This relationship suggests a potential route to inferring uncertainty in the

OSIRIS and CALIOP data during the SAGE II to SAGE III/ISS gap period by estimating

changes in the extinction coefficient slope (or Angstrom coefficient) based on perturbations in

those instruments' measured quantities. There is uncertainty to the details of this analysis,

particularly as it relates to the timing of the measurements following the eruption, thus the

apparent linearity of the 8 data points should be interpreted cautiously. Nonetheless, it should be possible for ESMs and GCMs with detailed aerosol microphysical models to calculate aerosol extinction coefficient at any wavelength and thus this analysis may provide the opportunity for a small-to-moderate volcanic plume closure experiment.

Despite the close timing of the two Ambae eruptions in 2018 eruptions (April and July), the
eruptions are clearly distinguishable in the SAGE III/ISS data shown in Figure 7f with the later eruption many times more intense than the earlier one (Kloss et al., 2020b). Individually, the Ambae (Vanuatu) eruptions in 2018 are similar to the Nevado del Ruiz eruption discussed in detail above as both show an increase in the extinction coefficient and extinction coefficient ratio relative to the values seen in early 2018 that is characteristic of most small-to-moderate
eruptions. However, the extinction coefficient ratio decreases following the second eruption suggesting that the second eruption may be an outlier to the generally observed behavior. To calculate the perturbations for these two events we use data from prior to the first eruption as the 'before' values for both though the results for the second eruption are insensitive to the perturbation caused by the earlier eruption. The initial Ambae eruption increased the extinction
coefficient ratio from 3.2 to 4.7 with an increase of 1020 nm extinction from about $10^{-4}$ to about $3 \cdot 10^{-3}$ km$^{-1}$. The second eruption initially increases the extinction coefficient ratio from 4.5 just prior to the eruption to 4.9 with the earliest observations shortly after the eruption that subsequently decrease to 4.1 when the aerosol extinction coefficient is a maximum. Aerosol extinction coefficient increases from $2.\times10^{-4}$ km$^{-1}$ to $1.3\times10^{-3}$ km$^{-1}$ or about a factor of 6 (Figure
7f). With these values, and despite appearances, both eruptions fit well with the majority of the other events (Figure 8b). In this case, the eruptions occur at slightly different altitudes so the apparent rise in the aerosol layer from the beginning to the end of the period is a little larger than

for most events (~2 km). In this case, particularly for the second eruption, the extinction change

is so large that the impact of the pre-eruption aerosol values is negligible.  Another interesting

feature is that the largest ratios after the eruption do not necessarily coincide with the largest

extinction. Figure 9 shows the extinction latitude/altitude cross sections for September 2018 for

521 nm (Figure 9a), 1020 nm (Figure 9b) and their ratio (Figure 9c).  It is clear here that the

maximum in the extinction ratio lies below the main peak in extinction coefficient in the tropics

and, notably stretches to higher southern latitudes and the maximum values actually occurs near

30° S despite more inhomogeneous conditions at this latitude than in the tropics. This is not an

obvious outcome, but it is consistent with the general observation that the largest perturbations in

extinction ratio occur with smaller extinction coefficient perturbations as shown in Figure 8b. It

also shows the importance of keeping in mind that the relationship between extinction coefficient

perturbation and overall extinction ratio in Figure 8b is for the densest part of the volcanic plume

and not all parts of the volcanic cloud. That the dependence of aerosol extinction coefficient

perturbation ratio on extinction coefficient perturbation occurs within a particular eruption as

well as among different eruptions (for the peak values shown in Figure 8) implies that a

consistent physical process is at work.

There are two events lying considerably away from Figure 8b's main curve: Kelut (1990) and

Ruang.  For Kelut, the first observations of the plume take place about 10 days after the eruption.

This is where the extinction ratio is the lowest (Figure 7a) and it increases from 2.2 to 2.6 in

following few weeks and then to 2.9 at the end of the observation period.  Ruang shows some

similar features with the low perturbation ratio (2.9) occurring shortly after the eruption followed

by a recovery toward larger values in the weeks that follow (3.9). The Kelut scatter plot (Figure

10) shows that while the scatter of extinction coefficient and ratio are compact for most of this

period, there are some observations of higher extinction and ratios approaching one which occur in the earliest observation period suggesting the immediate presence of large aerosol (>0.5 μm). While the data itself does not provide certainty, it is possible that an extinction-dominating presence of ash particles rather than sulfuric acid particles in the main aerosol layer immediately after the eruption may push its perturbation location below the rough curve suggested by most of the events. Similar data from Ruang is less illuminating due to a much smaller sample of data in the 50% duty cycle period of SAGE II data (after the end of 2000) and it is not possible to infer a cause for its anomalous position in Figure 8b. Both eruptions show increased aerosol extinction coefficient ratios away from the main aerosol peak suggesting, at least in part, a behavior more consistent with most eruptions.

Another interesting feature are differences between the Nevado del Ruiz, Cerro Hudson and Raikoke eruptions which cause very similar extinction coefficient perturbations but different perturbation extinction ratios. The position of Nevado del Ruiz in Figure 8b is consistent with the overall perturbation relationship. Raikoke lies on the same side as the Kelut and Ruang eruptions but, unlike Kelut, there is little evidence of a mix of increased extinction coefficient observations with small and large extinction ratios (large particles inferred to be ash but possibly other compositions) at the peak extinction level as essentially the data uniformly shows small extinction coefficient ratios following the mean relationship in Figure 7g. Since Raikoke is one of only two mid latitude eruptions in the data set, it is possible that latitude plays a role in the perturbation relationship. However, Cerro Hudson lies closer to Nevado del Ruiz's position and is a similar event to Raikoke as it occurs at a similar latitude (though opposite hemisphere) and season and at a similar pre-eruption aerosol extinction coefficient level. It is possible that atmospheric conditions or some detail of eruptions can have a modulating impact on how events

manifest themselves in extinction coefficient and ratio but not be easily detectable from the data

alone. For instance, for Raikoke, we cannot exclude the possibility of the presence of small

amounts of ash embedded in the main aerosol layer with the sulfuric acid aerosol influencing the

extinction coefficient and ratio. The presence of ash following the Raikoke eruption has been

inferred above 15 km and perhaps as high as 20 km (Muser et al., 2020;Kloss et al., 2020a). In

this case, it is possible that the ash is coated with sulfuric acid and these particles may freeze. It

is also possible that pyrocumulus events in Alberta, Canada and Siberia occurring around the

time of the Raikoke eruption (Yu et al., 2019), play a role in the evolution of extinction

following this event.   Overall, there is substantial opportunities for complex optical properties in

this eruption. To some extent, while we are fortunate to have as many events for this analysis as

we do, it is still a relatively small sample and some factors that can impact the extinction

coefficient/ratio relationship may not be fully revealed.

## 4 Discussion

Based on the observations discussed above, but without a detailed simulation of the aerosol

microphysical processes at play, we speculate that most small-to-moderate eruptions are initially

dominated by small (~1 nm), mostly homogeneously nucleated sulfuric acid particles that are

present in very large number densities (Deshler et al., 1992;Boulon et al., 2011;Sahyoun et al.,

2019). As shown in Figure 2a, due to their small size, these particles are initially extremely poor

scatterers and thus would not impact the SAGE-like extinction measurements. However, as they

coagulate into steadily larger particles (possibly also consuming small-sized aerosol present in

the pre-existing aerosol layer) and further condensation occurs, they would produce perturbations

to the observed aerosol extinction and ratio that reflect their magnitude. This process generally

causes an increase in aerosol extinction coefficient ratio but may produce the opposite effect

depending on the properties of the aerosol present prior to the eruption (which is discussed in more detail below). The coagulation process continues producing ever larger aerosol and smaller particle number densities until coagulation is no longer efficient at the time scales we examine

here and with respect to mixing of the material within the stratosphere. Some eruptions, like that of Raikoke in 2019 clearly depart from this conceptual model as we discuss further below. For large magnitude eruptions, like Mt. Pinatubo, it is possible that volcanic precursor gases and sulfuric acid vapor primarily condense onto existing aerosol and these, and very small homogeneously nucleated aerosol particles, rapidly (compared to the measurement frequency of

SAGE-like measurements) coagulate to form much larger-sized aerosol than after small-magnitude eruptions and, thus, the aerosol extinction coefficient ratio decreases extremely rapidly toward a value of 1. This alternative is not consistent with the observations of most small-to-moderate eruptions shown in Figure 8 and the conceptual model we describe below is not intended to capture this behavior.

To demonstrate how the homogeneous nucleation/coagulation process could impact SAGE-like observations, we have used a conceptual model that simulates a volcanic perturbation as single radii sulfuric acid particles that begin at 1 nm radius and grows to large particle sizes (500 nm) but hold the total volume of new aerosol material constant. The goal is to show that the large aerosol extinction coefficient perturbation ratios observed following small to moderate eruptions

are consistent with the presence of many small particles that grow through coagulation to larger particles with smaller extinction ratios. The model also shows why similar sized eruptions can appear differently in extinction coefficient measurements depending on the state of stratospheric aerosol prior to the eruption.  This is an extremely simple view of how the aerosol size changes after an eruption and cannot capture the details of the microphysical processes going on in the

volcanic aerosol layer, nonetheless, we believe that it provides a reasonable interpretation of the

observations and it provides a starting point for a model for post-volcanic aerosol spectral

dependence that could be useful for OSIRIS and similar measurements including a degree of

predictability for events not measured by SAGE instruments such as Sarychev, Kasatochi and

Nabro.  It may also be useful in comparisons of SAGE-like observations and results from GCMs

and ESMs.

For the model, we determine the volume density of aerosol required to produce 1020-nm

extinction coefficient perturbations of $10^{-4}$, $10^{-3}$, and $10^{-2}$ km$^{-1}$ at a single-radius of 500 nm. This

can be expressed using

$$n(r) = \frac{\delta k_\lambda}{Q_\lambda(r)\pi r^2} \qquad (3)$$

and

$$V = \frac{4\pi r^3 n(r)}{3} \qquad (4)$$

where $\delta k_\lambda$ is the extinction coefficient perturbation at wavelength $\lambda$ (in this case 1020 nm), $r$ is

perturbation particle radius (500 nm), $n(r)$ is the inferred perturbation particle number density,

$Q_\lambda(r)$ is the Mie extinction efficiency for the wavelength (shown for 525 and 1020 nm in Figure

2a) and radius considered for sulfuric acid aerosol at stratospheric temperatures, and $V$ is the

required volume density of aerosol. The choice of 500 nm for this calculation is somewhat

arbitrary and any value would not affect the conclusions drawn from this study. For an extinction

perturbation of $10^{-2}$ km$^{-1}$ the number density is 4.50 cm$^{-3}$ with a volume density of 2.37 $\mu$m$^3$/cm$^3$.

Holding V fixed, we compute number density and the aerosol extinction coefficient perturbation

as a function of radius at 525 and 1020 nm using

$$n(r) = \frac{3V}{4\pi r^3} \qquad (5)$$

and

$$\delta k_\lambda = Q_\lambda(r)n(r)\pi r^2 \qquad (6)$$

for radii, $r$, from 1 to 500 nm. The ratio of these extinction coefficient perturbations follows the

relationship shown in Figure 2b. Finally, we add 'before' aerosol extinction coefficient values

we previously determined for the Nevado del Ruiz eruption and the July 2018 Ambae eruption

and show these relationships in Figure 11a and 11c respectively.  Due to their different pre-

eruption extinction levels, the extinction ratio plots shown for the two volcanic events are

notably different despite having identical extinction coefficient perturbations at 525 and 1020 nm

computed using the above relationships. This is consistent with the data shown in Figure 8a. To

some extent, the radius axis in this plot is akin to a time axis though a particularly non-linear one.

It is likely that the transition across the smallest size particles is extremely rapid (relative to

SAGE-like observation timescales at least) and the large end of the timescale may effectively be

reached rapidly for large events like Mt. Pinatubo but effectively never for small-to moderate

eruptions due to the other processes that control coagulation and other aspects of aerosol

morphology. Indeed, the first observations of the main Mt. Pinatubo cloud in early July 1991, a

few weeks after the eruption, show an extinction coefficient ratio of essentially 1. Whether this

would have been the case with observations on say immediately after the eruption is an

interesting unknown. In the aftermath of the second Ambae eruption, as shown in Figure 7f, the

aerosol extinction coefficient ratio maximum occurs before the maximum in extinction at 1020

nm and in fact, the ratio has decreased by the time extinction coefficient at 1020 nm is a

maximum. This is reproduced by the model for the 'Ambae' eruption where the maximum in

aerosol extinction ratio is observed at significantly smaller radii (Figure 11a) than for which the 1020-nm aerosol extinction coefficient is a maximum (Figure 11b). This behavior is also

exhibited in the model for Nevado del Ruiz eruption the aerosol extinction coefficient perturbation ratio (shown in Figure 11c) is not as peaked it nonetheless clearly reaches a maximum at smaller radii than where 1020-nm  aerosol extinction coefficient reaches a maximum (shown in Figure 11d).

If the initial growth to 200 nm is rapid at SAGE temporal sampling scales (~monthly), the model

simulations qualitatively reproduce the increase in extinction coefficient ratio seen in many of the eruptions analyzed with a step increase in extinction coefficient ratio followed by a decrease in time. In addition, these results show that, while the extinction coefficient perturbations themselves may be insensitive to the 'before' stratospheric state, the result is not. In fact, scenarios can be easily constructed in which the same eruption, again with minimal interaction

with the preexisting aerosol, results in a different sign in the slope of the change in extinction coefficient ratio. Obviously, we must exercise caution in interpreting the observations based on the simple model employed here. For instance, since we do not know the timescale of coagulation, significant uncertainty remains in how to interpret Figure 8b in a temporal sense. Moreover, aerosol volume density is unlikely to be constant over this time as the conversion of

$SO_2$ to $H_2SO_4$ has a time constant on the order of 30 days and depends on the magnitude of the eruption. Nonetheless, while not a primary goal for this study, we argue this very simple model suggests that SAGE II/III observations are consistent with volcanic material primarily nucleating homogeneously followed by coagulation whose timescale depends on the magnitude of the eruption. In the end, however, only through closure experiments between observations such as

these and GCMs and ESMs with detailed microphysical models can certainty be obtained.

**5 Conclusions**

Herein, we have used SAGE II/III observations to examine the behavior of stratospheric aerosol extinction coefficient in the aftermath of small-to-large magnitude volcanic events with a primary goal of understanding how these events manifest themselves in SAGE-like observations.

We have focused on the initial plume development at the peak extinction levels and not the long-term development or the details of its distribution as transport and other aerosol processes such as sedimentation have not been considered. We have found that observations of the impact of volcanic eruptions on stratospheric aerosol as measured by the SAGE series of instruments show at the peak extinction levels, under most circumstances, a crude independence to the

characteristics of the preexisting aerosol and a correlation between the magnitude of the enhancement in aerosol extinction coefficient and its wavelength dependence as shown in Figure 8b. While this relationship is insensitive to the preexisting aerosol level, the preexisting aerosol can modulate the observed changes in aerosol extinction coefficient ratio. The analysis is straightforward for tropical eruptions but more challenging for mid and high latitude eruptions

where transport is generally more complex than in the tropics.  Also, it is possible that volcanic events with significant amounts of ash may behave considerably different than those dominated by the sulfuric acid component.

The perturbation relationship, shown in Figure 8b, is based only on the measurements themselves and makes no assumptions about the underlying composition or size distribution of

the aerosol.  In this respect, it is a unique tool to intercompare observations and interactive aerosol models used in GCMs and ESMs. This should be extremely straightforward as extinction coefficients can be calculated from aerosol products already produced by these modules though care would need to be exercised to reproduce the observations used herein. Since the results span

a large dynamic range of aerosol extinction coefficient perturbations (> two orders of

magnitude), the testing range covers a significant range of volcanic events.  Since the observed

relationship is well behaved, testing is potentially not limited to observed volcanic events but

may be applied to hypothetical events or historical events for which space-based observations do

not exist.

A longer term goal is to assess data quality of data sets consisting of a single wavelength

measurement of aerosol extinction coefficient or similar parameter particularly when a fixed

aerosol size distribution is a part of the retrieval process. This is important as a part of the data

quality assessment of these data sets as well as their use in long-term data sets such as GloSSAC.

In this regard, the results are mixed. It is clear from Figure 8b that the wavelength dependence of

a predominating sulfuric acid volcanic event can be estimated from the relationship shown

therein. Since a fixed particle size distribution is used in the OSIRIS retrieval process, a fixed

wavelength dependence is effectively intrinsic to the OSIRIS aerosol extinction coefficient

retrieval process. The use of these results in OSIRIS retrievals is an on-going study which we

hope will result in positive improvements in the OSIRIS aerosol data products in the future.  In

the short term, we believe that we may be able to use these results in spot applications such as

assessing the extinction error due to the fixed aerosol size distribution in the immediate aftermath

of an event.

**Code and data availability.**

SAGE    II    (https://doi.org/10.5067/ERBS/SAGEII/SOLAR_BINARY_L2-V7.0) and SAGE

III/ISS data (https://doi.org/10.5067/ISS/SAGEIII/SOLAR_HDF4_L2-V5.1 ) are accessible at

the NASA Atmospheric Sciences Data Center. GloSSAC v2.0

(http://doi.org/10.5067/GLOSSAC-L3-V2.0 ) is available from the same location. Data analysis products shown herein are available from the corresponding author.

**Author contributions.**

LWT developed the analysis tools used throughout the paper and was the primary of author of

the manuscript. MK and LR advised the author particularly in relationship to the GloSSAC data set and issues related to OSIRIS data quality and algorithms. AS suggested the conceptual model used to characterize the way small-to-moderate volcanic eruptions affect aerosol extinction ratio. CvS and TNK advised regarding the use and modeling of the SAGE data sets. Finally, all authors provided substantial input on the construction of the manuscript and figures.

**Competing interests.**

The authors declare that they have no conflict of interest.

**Financial support.**

LWT, MK and TNK are supported by NASA's Earth Science Division as a part of the ongoing development, production, assessment, and analysis of SAGE data sets. Stratospheric aerosol

research at the University of Greifswald (CvS) is funded by DFG (project VolARC of the DFG Research Unit VolImpact, FOR 2820; grant number 398006378). AS received funding 355 from UK Natural Environment Research Council (NERC) grants NE/S000887/1 (Vol-Clim) and NE/S00436X/1 (V-PLUS).Work performed by LR was funded by the Canadian Space Agency under the Earth Science System Data Analyses program.


**Acknowledgements.**

We acknowledge the support of NASA Science Mission Directorate and the SAGE III/ISS mission team. We would like to thank reviewer Pasquale Sellitto and two anonymous reviewers for their contributions to this manuscript.


Table 1. Volcanic eruptions and smoke events that significantly impact stratospheric aerosol levels in the Version 2.0 of the GloSSAC data set (Kovilakam et al., 2020) and denoted in Figure 1 using the abbreviation in brackets following the name.

| Volcano Name | Eruption Date | Latitude |
|---|---|---|
| St. Helens (He) | 27 Mar 1980 | 46° N |
| El Chichon (El) | 4 Apr 1982 | 17° N |
| Nevado del Ruiz (Ne) | 14 Nov 1985 | 5° S |
| Kelut (Ke) | 10 Feb 1990 | 8° S |
| Pinatubo (Pi) | 15 Jun 1991 | 15° N |
| Cerro Hudson (Ce) | 12 Aug 1991 | 46° S |
| Rabaul (Ra) | 19 Sept 1994 | 4° S |
| Ruang (Rn) | 25 Sept 2002 | 2° N |
| Manam (Mn) | 27 Jan 2005 | 4° S |
| Soufriere Hills (Sh) | 20 May 2006 | 16° N |
| Tavurvur (Tv) | 07 Oct 2006 | 4° S |
| Chaiten (Ch) | 02 May 2008 | 42° S |
| Okmok (Ok) | 12 Jul 2008 | 55° N |
| Kasatochi (Ka) | 07 Aug 2008 | 55° N |
| Fire/Victoria (Vi) | 07 Feb 2009 | 37° S |
| Sarychev (Sv) | 12 Jun 2009 | 48° N |
| Nabro (Nb) | 13 Jun 2011 | 13° N |
| Kelut (Ke) | 13 Feb 2014 | 8° S |
| Calbuco (Cb) | 22 April 2015 | 41° S |
| Canadian Wildfires (Cw)[1] | August 2018 | 51° N |
| Ambae (Am) | 5-6 April 2018/27 July 2018 | 15° S |


[1] *Canadian Wildfire (Cw) occurred in August 2017 created pyrocumulonimbus (PyroCb) that injected smoke into the stratosphere (Peterson et al., 2018). This event is also marked in Figure 1.

Table 2.  Volcanic events observable in the SAGE II (1984-2005) and SAGE III/ISS (2017-present) records including the total number of observations used in the analysis.

| Eruption | Date | Latitude | Altitude (km) | SAGE Observations | Julian Date of Eruption(s) |
|---|---|---|---|---|---|
| Nevado del Ruiz | 13 November 1985 | 5° N | 20.5 | 634 | 317 |
| Kelut | 10 February 1990 | 8° S | 20.5 | 523 | 41 |
| Mt. Pinatubo | 17 June 1991 | 15° N | 22.0 | 433 | 168 |
| Cerro Hudson | 8 August 1991 | 46° S | 11.5 | 1162 | 221 |
| Ruang | 25 September 2002 | 9° S | 18.5 | 255 | 268 |
| Manam | 27 January 2005 | 4° S | 20.0 | 219 | 27 |
| Ambae | 5-6 April 2018/28 July 2018 | 15° S | 18.0 | 858 | 95/209 |
| Raikoke | 22 June 2019 | 48° N | 15.0 | 1014 | 173 |
| Ulawun | 26 June 2019/3 August 2019 | 5° S | 18.5 | 491 | 177/215 |


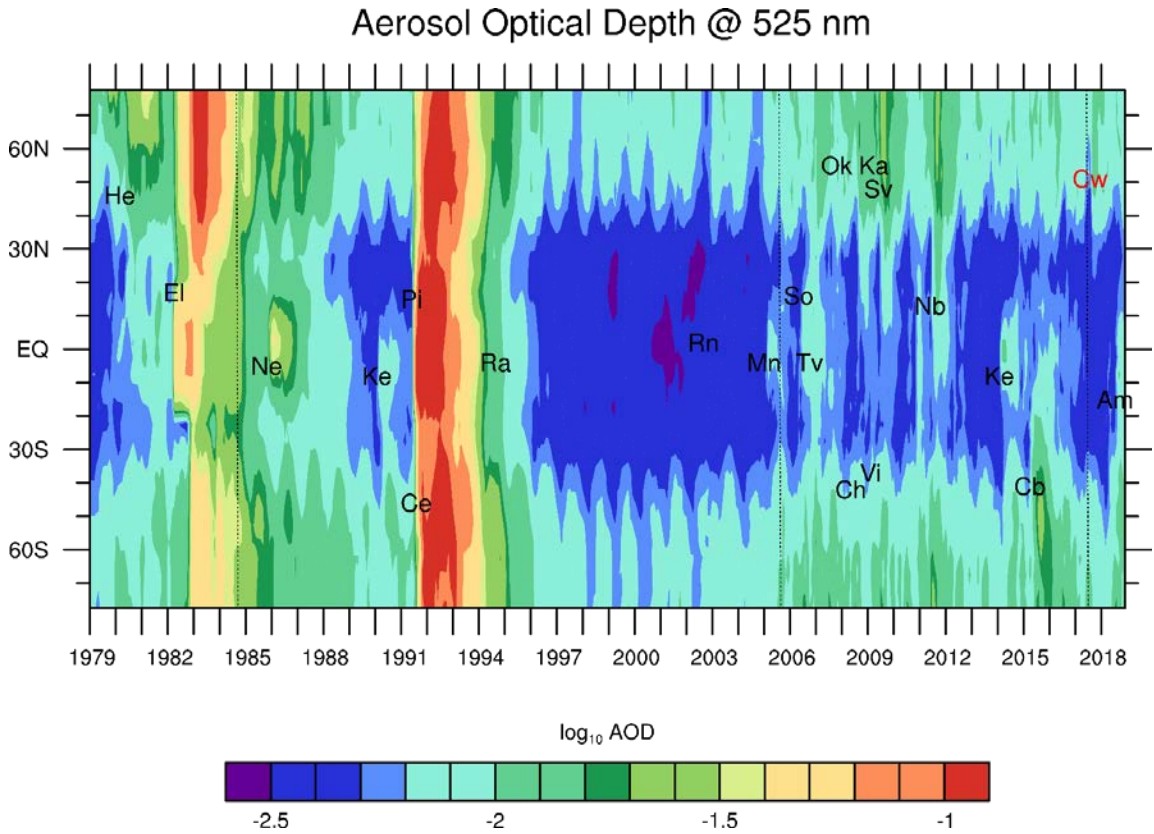

Figure 1. Stratospheric aerosol optical depth at 525 nm from GloSSAC v2.0 [Kovilakam et al., 2020]. Volcanic and similar events are denoted using symbols given in Table 1. Dotted vertical lines indicate (from left to right) the start of the SAGE II mission in 1984, the end of the SAGE II mission in 2005, and the start of the SAGE III mission in 2017.


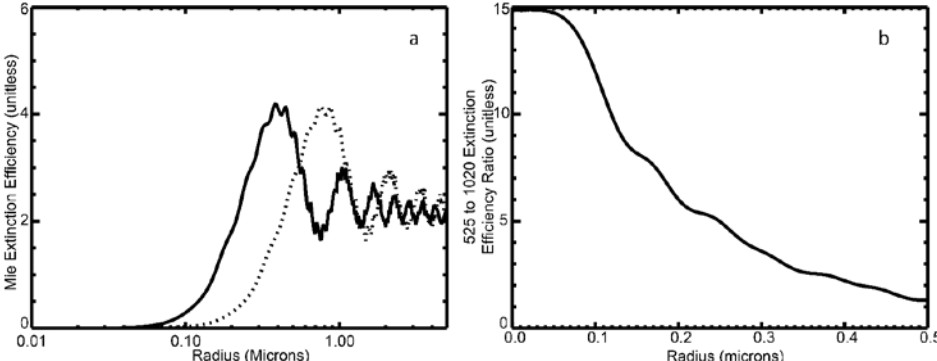

Figure 2. (a) Mie extinction efficiency for sulfuric acid droplets at stratospheric temperatures at 525 (solid) and 1020 nm (dashed). (b) The ratio of extinction coefficient at 525 to 1020 nm for single particles as a function of radius for sulfuric acid aerosol at stratospheric temperatures.

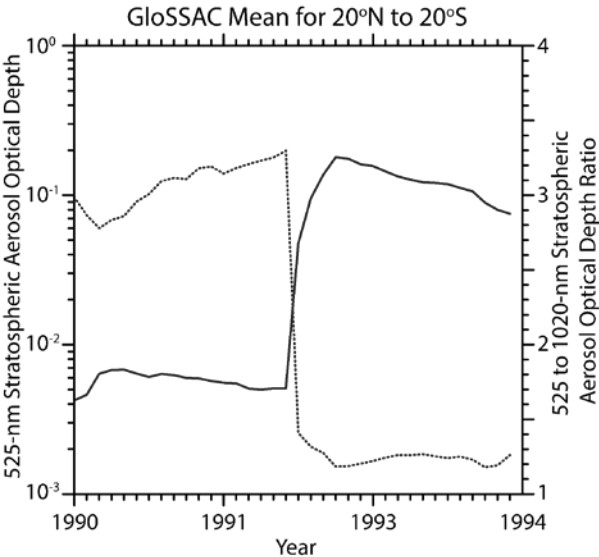

Figure 3. The GloSSAC v2.0 depiction of 525-nm aerosol optical depth (solid) and 525 to 1020-nm stratospheric aerosol optical depth ratio (dotted) for 1990 through the end of 1993 encompassing the Kelut eruption in early 1990 and the Mt. Pinatubo eruption in mid-1991.

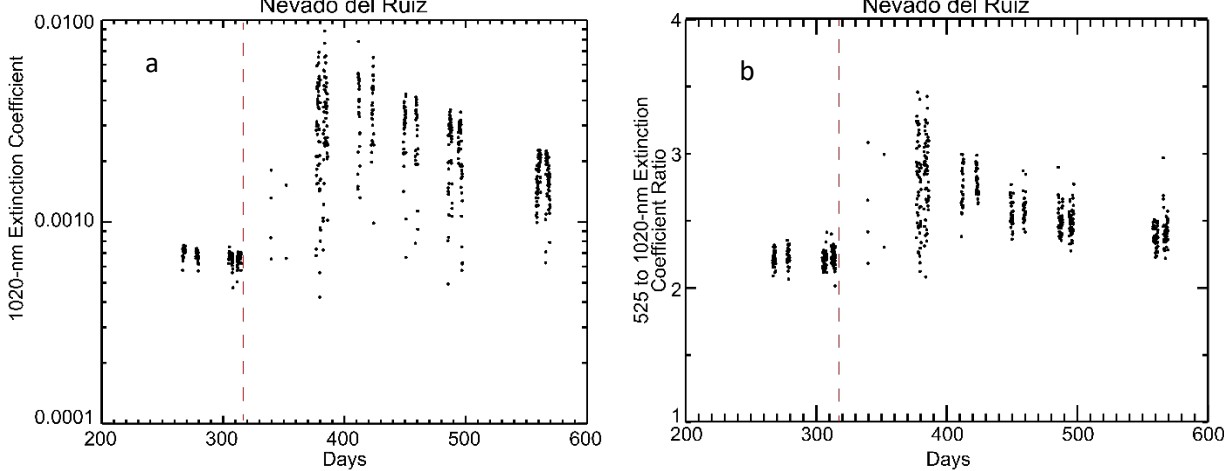

Figure 4. The time series of SAGE II 1020-nm aerosol extinction coefficient in km$^{-1}$ (a) and 525 to 1020-nm aerosol extinction coefficient ratio (b) at 20.5 km between 10S and 10N in days from 1 January 1985 (Day 1) thus the first day is 19 July 1985, the eruption occurs on day 317 (13 November 1985), and the plot ends on 23 August 1986. The date of the eruption is denoted by a vertical dashed red line.

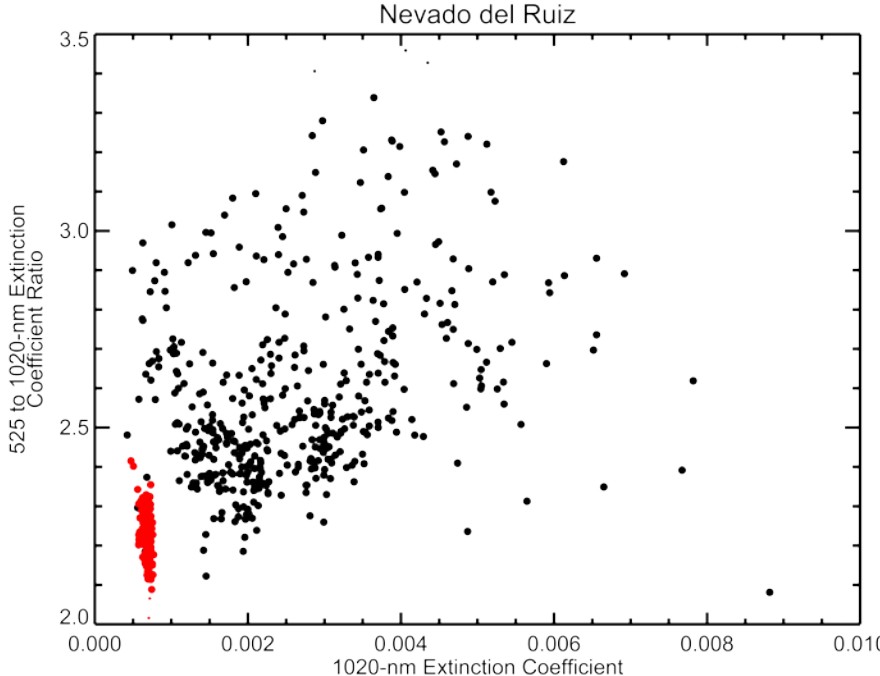

Figure 5. Same data as shown in Figure 4a and 4b except now plotted as 1020-nm aerosol extinction coefficient (in km[-1]) versus the extinction coefficient ratio. The extinction coefficient ratio is a rough estimate of the size of aerosol particles that dominate extinction. Values near 1 suggest particle radius greater than ~0.4 μm with increasing value indicating smaller particles. Values for observations prior to the eruption are red. All data is for 20.5 km.




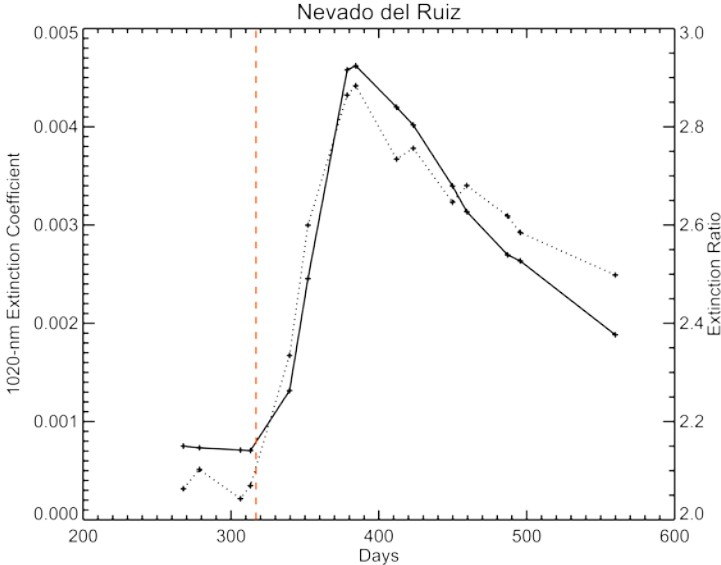

Figure 6. Same data as shown in Figure 4 except averaged in temporal data clusters. In this figure, extinction coefficient is the solid line and the extinction coefficient ratio is the dotted line. The date of the eruption is denoted by the vertical red dashed line.

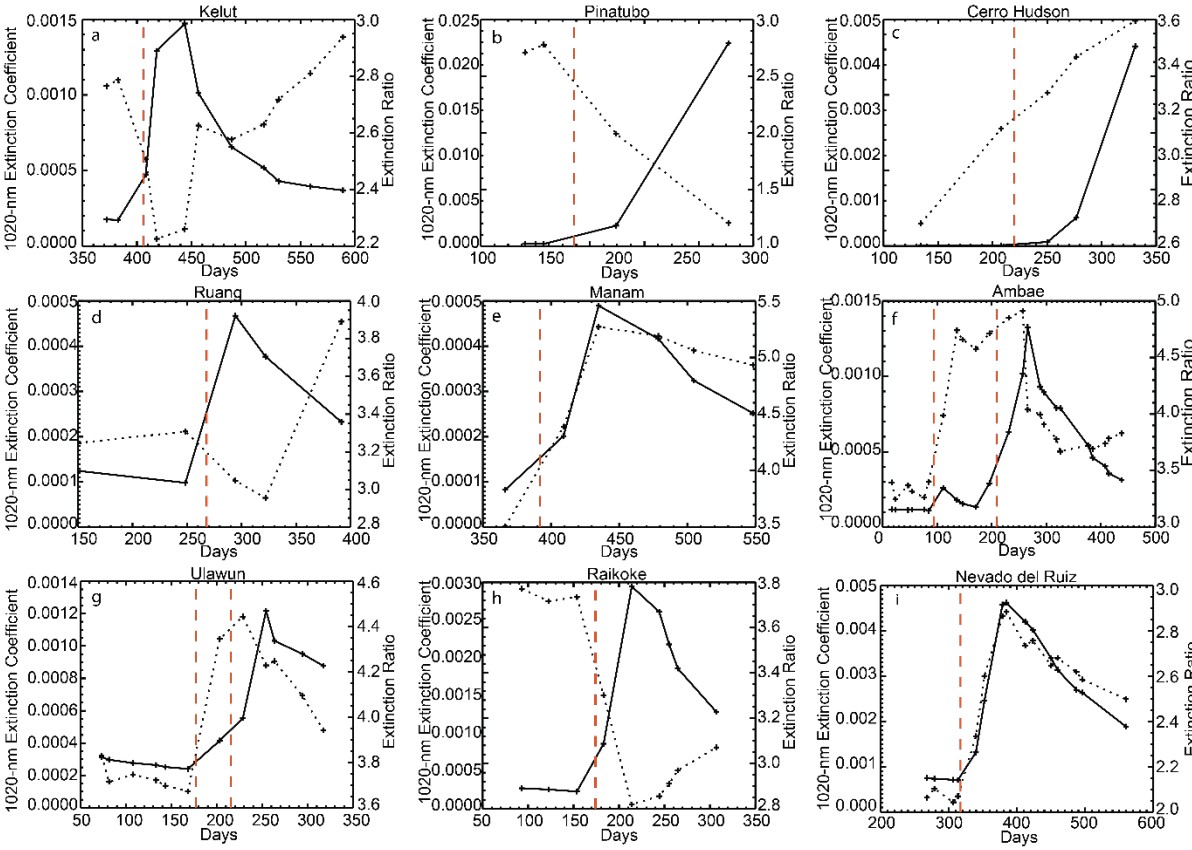

Figure 7. Similar analysis as shown in Figure 6 except for Kelut in 1990 (a), Mt. Pinatubo (b) and Cerro Hudson (c) in 1991, Ruang in 2002 (d), Manam in 2005 (e), Ambae in 2018 (f), Ulawun (g) and Raikoke (h) in 2019. In each frame, extinction coefficient is the solid line and the extinction coefficient is the dotted line. The dates of the eruptions are denoted by the vertical red dashed lines. The plot for the Nevado del Ruiz eruption shown in Figure 6 is repeated here as frame (i) for comparative purposes. Days refer to the number days since the start of year in which the analysis begins for an individual eruption. For figures (a) to (i) these years are 1989, 1991, 1991, 2002, 2004, 2018, 2019, and 2019, respectively.


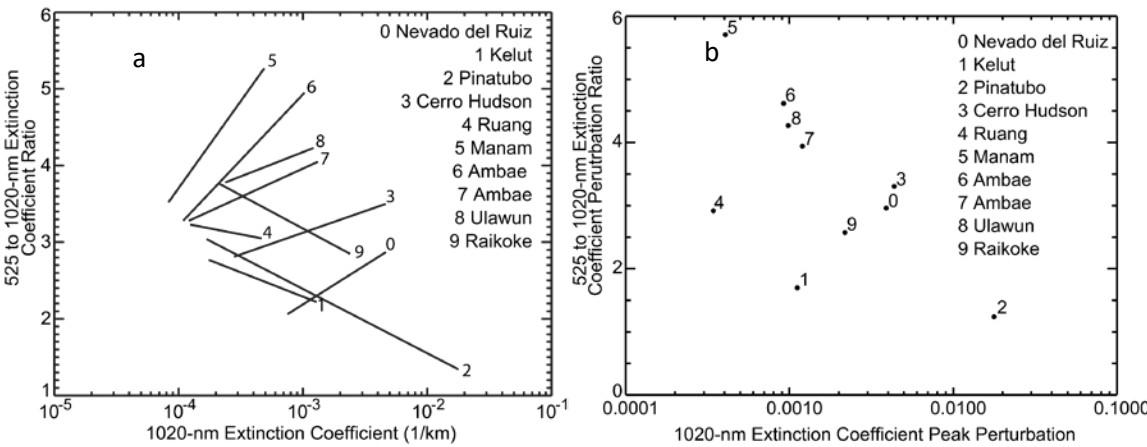

Figure 8. The before (left-hand) to peak 1020-nm aerosol extinction coefficient (right-hand point) for the 10 eruptions considered in this study is shown in frame (a) with the differences between them (perturbations) are shown in frame (b).



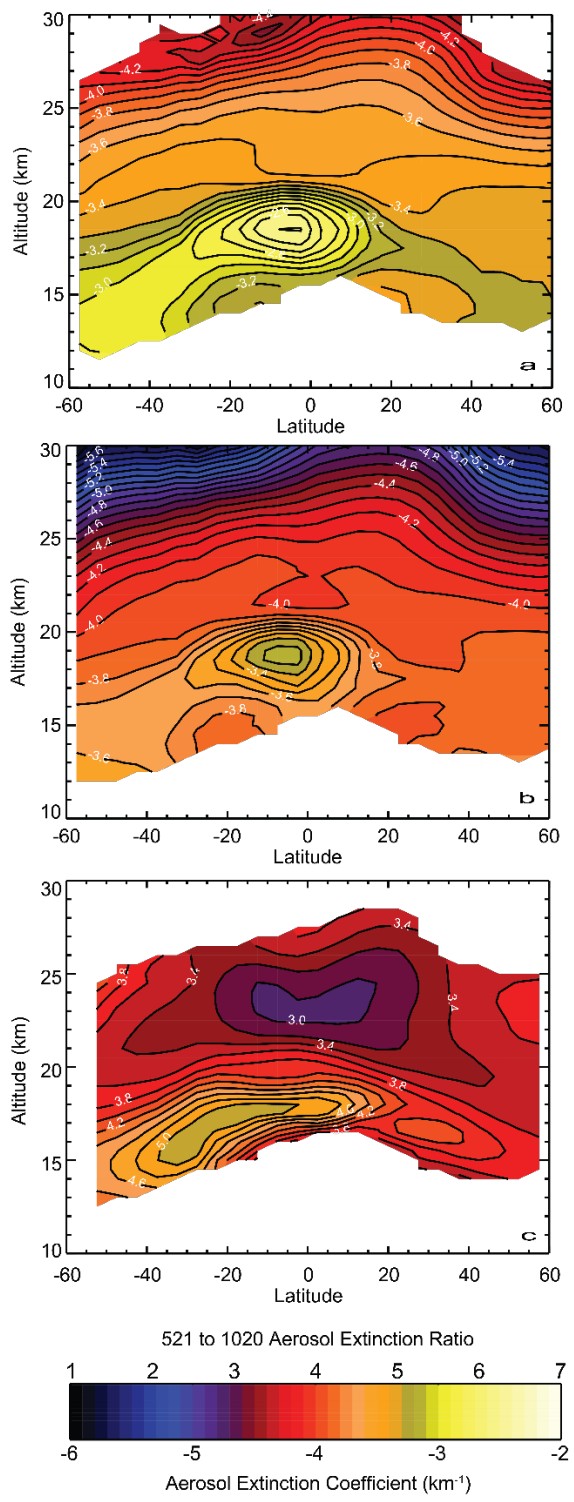

Figure 9. Mean SAGE III/ISS 525 (a) and 1020 nm (b) aerosol extinction coefficient and 525 to 1020-nm aerosol extinction coefficient ratio (c) as a function of latitude and altitude from September 2019 shortly after the second 2019 eruption of Ambae (July 2019; 15°S).

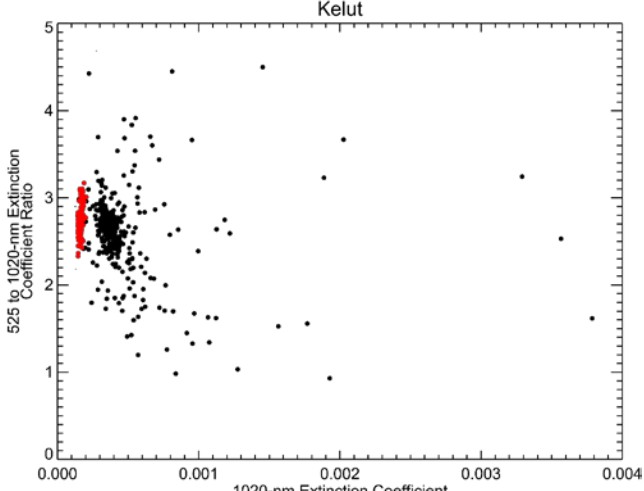

Figure 10. SAGE II 525 to 1020 nm aerosol extinction coefficient ratio plotted versus 1020-nm aerosol extinction coefficient in km$^{-1}$ during the Kelut event from December 1989 through August 1990 plotted at 20.5 km between 20S and the Equator. Measurements occurring before the eruption are colored red.

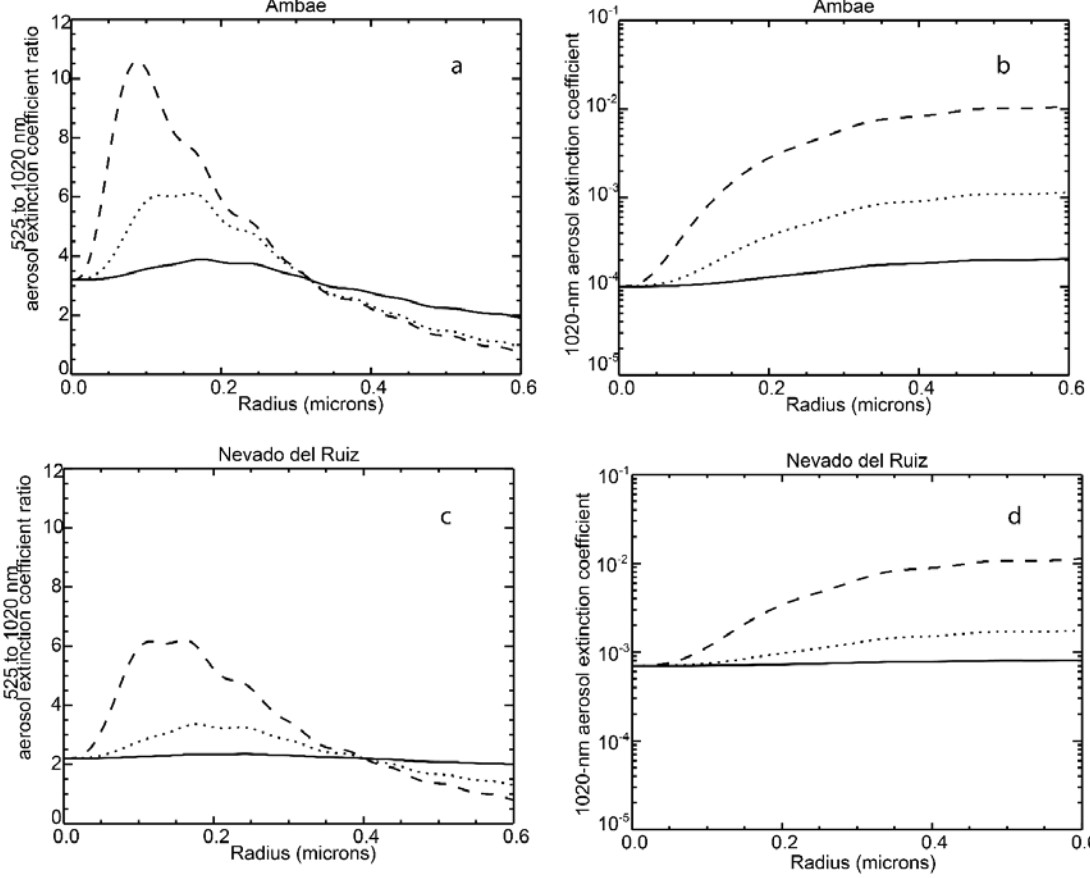

Figure 11. Estimated 525 to 1020-nm aerosol extinction ratio and 1020-nm aerosol extinction coefficient for the second Ambae eruption (a and c) and Nevado del Ruiz (b and c) computed using fixed aerosol volume density perturbations and single-radii particles that yield an extinction coefficient perturbation at 525 nm of $10^{-4}$ (solid), $10^{-3}$ (dotted), and $10^{-2}$ $km^{-1}$ (dashed) using rough 'before' 525 and 1020 nm extinction coefficient values for each eruption.

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
