# Peer review of "Evidence for the predictability of changes in the stratospheric aerosol size following volcanic eruptions of diverse magnitudes using space-based instruments"

_Atmospheric Chemistry and Physics, 2020_

## Referee Comment (RC1) · Pasquale Sellitto (Referee) · 3 Jul 2020

The manuscript "Evidence for the predictability of changes in the stratospheric aerosol size following volcanic eruptions of diverse magnitudes using space-based instruments, Thomason et al." discusses the relationship of multiwavelength aerosol extinction observations, in the first phases of volcanic plumes dispersion following small-to-strong stratospheric eruption, and the apparent size of the particles in the plume. From my perspective, the main result of this work is the evidence of a clear relationship of the extinction ratio (shorter-to-longer wavelength, namely 525/1020 nm, hereafter referred

to as ER) and the strength of the eruption in terms of the aerosol extinction at 1020 nm (hereafter referred to as AE), see Fig. 8b. This relationship is associated to the apparent particles size in the aerosol layer (Fig. 2). Even if this behaviour is probably expected, I personally think that a systematic study of this relationship is very interesting and important. The identification of volcanic plumes by means of the concurrent increases of the aerosol extinction (or the integrated optical depth) and modifications of the spectral variability of this extinction (or the Ångström exponent) has been exploited, in the past, in different studies – and it is a tool that I personally use a lot. Nevertheless, a systematic effort to: a) study this aspect over a long observation series, or even b) construct a theoretical basis, has never been attempted, to my knowledge. Thus, I think that such kind of work would deserve immediate attention and rapid publication. Unfortunately, the present manuscript, while having all elements to provide the community with both points a and b mentioned above, is somewhat flawed in two aspects, that I mention in the following. I encourage the Authors to tackle these two "major" issues, as well as a number of specific issues that I also list in the following, and I'll be happy to re-evaluate the manuscript revised accordingly, once the due modifications are done.

Sincerely,

Pasquale Sellitto

*Major comments:*

1) The conceptual model defined to connect the apparent radius of the aerosol layer and AE/ER, as defined at L271-273 and shown in Figs. 2 and 11, is not completely clear to me. The model is defined for what looks like a monodispersed aerosol layer (which is also mentioned at L102), which is fine when discussing this in theory (Fig. 2) but is a little bit odd when applying to real data (Fig. 11). In the real world, the size distribution would not be monodispersed, so I guess a more realistic size distribution should be used, which is still not very complicated using a mono-modal size distribution

with varying mean radii. Another thing that gets me confused is that in the equation at L271, it looks to me that the numerator of the ratio on the right is not the perturbation delta_k but rather k (otherwise, you don't have number density n(r) but the perturbation in number density delta_n(r)). Like this, it looks like you're accounting twice for the background, when you later scale the result with respect to the background. And also, the Mie scattering efficiency is calculated with a Mie code (which one?) and is based on an assumption of the particles' composition (their refractive index): which is your assumption? If, as I imagine, the Authors have supposed a pure sulphates plume, why the ash has been neglected – it may be important in the first phases of some eruptions, e.g. Raikoke and Kelud? And again, if it is sulphates, what has been supposed for the mixing ratio of sulphuric acid in the sulphate aerosols droplets, which is a factor that can modulate the extinction of the particles, at least at longer wavelengths? By using the Mie scattering coefficient, you suppose that the absorption can be completely neglected. If ash is to be considered, this might not be completely true. All these aspects have to be clarified or maybe the model has to be slightly refined to account for the mentioned problems.

2) The manuscript structure has to be improved. In the present version, the main results and the overarching narrative are not completely clear. First, the Introduction fails to present the motivations of this work. Some elements of motivation are in Sect. 2 rather than in the Introduction. It is stated two times that "The primary goal of this effort was to assess data quality of data sets consisting of a single wavelength measurement of aerosol extinction coefficient or similar parameter particularly when a fixed aerosol size distribution is a part of the retrieval process." but I cannot see where this is discussed in the text. As stated in my introduction, I would rather say that the main motivation for this study (and it is an important motivation!) is to develop a means to identify volcanic plumes and to classify them based on the eruption strength, and I suggest mentioning this as a motivation. Section 2 does not present satisfactorily the SAGEII-III datasets and a lot of information is lacking. Care should be taken, throughout the manuscript, to introduce the Figures sequentially in the text and not to discuss

them before presenting their content. I add some specific points in the following Specific Comments.

*Specific comments:*

1) I don't understand the sense of the word "predictability" in the title, as you attempt no predictions

2) L15-16: there is a lacking mention to the ER, which is the measured parameter that actually is studied and vary for different eruptions strengths (and AE)

3) L16: "The relationship is measurement-based and does not rely on assumptions about the aerosol size distribution.": strictly speaking there is an assumption of monodispersed aerosol layer

4) L18-22: "Despite this limitation...particle size": these two points (the use in evaluating global models and the improvement of mono-spectral AE observations) is not discussed in depth in the manuscript so it is strange that it is mentioned in the Abstract
5) As stated at Major Comment #2, I feel that the Introduction is very synthetic and does not make a good job in motivating this work

6) L24: "Eruptions of volcanoes" –> "Volcanic eruptions"

7) L25: "Volcanically-derived aerosol": Here the you talk in general of volcanically derived aerosol and, later in the text, the discussion specialises on sulphates. A line is probably lacking here on the mention of the possible variety of volcanic particles (ash and sulphates)

8) L29: I think that Tang et al. 2013 do not discuss of the impact of volcanic aerosol on transport but of the decrease of ozone in the stratosphere and then, consequently, in the troposphere by stratosphere-troposphere exchanges. As one lines above you discuss about the radiative heating of volcanic aerosols, one can erroneously think that Tang et al. talk about radiative-dynamical interactions and the crossing of tropopause by plume self-lifting, which is not the case. Please correct

9) L31: "...either the measurements...": please develop a bit to clarify how measurements have been used in global models to derive climate impact of volcanic eruptions

10) Why not more classic section titles: "2. Data and Methods" and "3. Results"?

11) L45: "well-known SAGEII": it is well-known but it might be less known for a part of the readership of ACP. Thus, please suppress "well-known" and mention the years and months of operations of SAGEII, from launch to end of mission.

12) In Tab. 1 there are more eruptions than what studied in Sect. 3 and 4 and there are fires as well. While later in the text it is said that fires are not in the scopes of this work (while it would have been interesting to see where the points in Fig 8b locate for Canadian and Australian fires...), it is not clear to me why many eruptions possibly present in the datasets have not been included in this study: Sarychev, Kasatochi and Nabro are largely considered "major moderate eruptions" for their impact on the stratosphere but are neglected in this study

13) L55: Please briefly introduce this GloSSAC dataset. Also please mention in the corresponding reference that the relative manuscript is presently under discussion/review and add a link for the discussion paper

14) L59: "SAGEII record": here is very clear that mentioning the start-end of SAGEII operations is important

15) L61: "...subtly modulate climate...": probably it can be mentioned that the aggregated impact of these "small-to-moderate" eruptions is significant (see also Ridley, D. A., et al. (2014), Total volcanic stratospheric aerosol optical depths and implications for global climate change, Geophys. Res. Lett., 41, 7763– 7769, doi:10.1002/2014GL061541)

16) L63: as for SAGEII, the precise period of operations of SAGEIII/ISS should be clearly written

17) Fig. 1: the periods of operations (start/end of missions) of SAGEII and III should

be probably indicated in this figure, e.g. with vertical dotted lines. That would help a lot in the understanding of the different discussions of Sect. 2

18) L67: "Raikoke...2020): why not arranging these eruptions in chronological order?

19) L69: in the following paper about Canadian fires 2017, a similar method as the one discussed in this manuscript is used to identify and separate a fire plume from an anthropogenic plume (see their Fig. 2b): Kloss et al. Transport of the 2017 Canadian wildfire plume to the tropics via the Asian monsoon circulation, Atmos. Chem. Phys., 19, 13547–13567, https://doi.org/10.5194/acp-19-13547-2019, 2019.

20) L69: please note this pre-print manuscript on the Australian fires 2019-20: https://arxiv.org/abs/2006.07284

21) L71-72: "...a qualitative difference...": please mention a difference with respect to what? (That would be clearer if the periods of operations of SAGEII and III are indicated in Fig. 1)

22) L73-80: These motivations should be moved to the Introduction

23) L81-93: also more appropriate in the Introduction?

24) L95: what do you mean with "robust"?

25) Are footnotes allowed in ACP format?

26) L101-102: see Major Comment #2

27) L124-125: "These begin...Table 2)": Is this line to be moved earlier (e.g. L120)?

28) L129: "...O4 absorption...version.": Is there a reference for this underestimation? And also, a few words should be included to clarify why using the interpolation between 448 and 756 nm limits (or avoids) this underestimation: O4 has no absorption at these wavelengths and has a significant absorption at 521 nm?

29) L131: "...(602 nm... 521 nm)": why? (see previous comment)

30) L138: "...November 13...": please mention the year

31) 142: "The opposite of what..." → "This is the opposite of what..."

32) L143: "The extinction ratio becomes...": here talking about Nevado del Ruiz (not Pinatubo)?

33) Why not aggregating Fig 4 and Fig 5 in a unique figure with 3 panels?

34) Fig. 4: Why not restricting the yaxis scale to something like 2 to 4? There are no values <2 or >4.

35) Fig. 4 caption: What do you mean with "the scatter"? I would just say "the time series of 1020... and 525 to 1020..."

36) Fig. 5: why not evidencing the pre-eruptive points (the cluster for smaller values of the 1020-nm AE) with a different colour or different symbol? Can it be possible to identify the points in the earlier stage of eruption, as well (so to corroborate the hypothesis of a different cluster of values, i.e. with ash)?

37) L148-149: "The distinction...recognizable": this can put more in evidence in Fig. 5 (see previous comment

38) L150: Has a sulphuric acid hypothesis been proposed earlier? A precise point where ash is discarded is not present in the previous text

39) L151-152: "Generally,...events": Can this be shown more clearly?

40) L153-156: "This was particularly...higher latitudes events": all this part is not very clear to me

41) L193-194: "At some point...reasonable": this makes reference to Fig. 2? Please clarify

42) L197-201: "This relationship...space-based instruments": how this can be done (inferring uncertainty in mono-spectral observations and evaluating aerosol modules in

GCMs)? I feel that this should be discussed much more in depth

43) L202-: in general, in Section 3 and 4 discussions of volcanic events considering existing information in the literature are lacking. For Ambae, for example, in the paper already cited, Kloss et al., 2020 (by the way , please correct the reference as this paper is now published in JGR and no more in preprint), the plumes are detected using SAGEIII observations and simultaneous increases of the AE and the partial column Ångström exponent (Fig. 8 of Kloss et al.). This can be easily put in connection with Fig. 7f.

44) L226: "...Ruang...": please add year of eruption

45) L230-232: "The Kelud...um)": here is an example where your work can be put in context with existing literature. In the following paper, it has been shown that ash was present for a long time in Kelud plume: Vernier, J.-P, Fairlie, T. D., Deshler, T., Natarajan, M., Knepp, T., Foster, K., Wienhold, F. G., Bedka, K. M., Thomason, L., and Trepte, C. (2016), In situ and space-based observations of the Kelud volcanic plume: The persistence of ash in the lower stratosphere, J. Geophys. Res. Atmos., 121, 11,104– 11,118, doi:10.1002/2016JD025344.

46) Fig. 7: it would be useful to have the indications (red dashed lines) for the two eruptions of Ambae and Ulawun

47) Fig. 7 caption: the mention to the volcanoes names is probably redundant, as the volcanoes are also mentioned in the panels. Also "...vertical dashed lines." –> "...vertical red dashed lines."

48) Fig. 8b: Why not quantifying this trend (linear regression and correlation parameter)?

49) L242: "...(possible ash)...": but also sulphate-coated ash or large sulphates in the accumulation mode possible, how do you exclude these?

50) L248-250: "For instance, for Raikoke...and ratio.": this makes reference to the

(complicated) issue of the mixing state of the aerosol population, including the possibility that ash is sulphate-coated and/or these particles may freeze. Probably a discussion about that is needed here.

51) L250-251: "It is also possible...this event.": interesting, can you please develop this point a bit?

52) L257: "These are initially...Figure 11)": How is it visible in Fig. 11. And also, why Fig. 11 is discussed before its content is defined (in the following lines)?

53) L257: "...but coagulate...": if talking about coagulation, why not of heterogeneous nucleation/condensation over pre-existing particles (sulphate aerosol or ash)? At this point, it looks clear to me that a discussion on the mixing state and aerosol microphysics is quite needed

54) Equations at L270-273 and inherent discussion: see Major Comment #1

55) Please add equation numbering

56) The priority of argumentations in the Conclusions is not clear to me. The main results of this work are probably the evidence of the dependence of the ER from the eruption intensity and AE (Fig 8b) but this is not even mentioned in the Conclusions

57) L327-328: "The primary goal...process": as mentioned above (Major Comment #2 and Minor Comment #4) this is not discussed in the text, so it is strange to see this in the Conclusions

58) L330-331: "It is clear...therein": this is actually not very clear to me: in the text there is no assumption on the aerosol chemical composition.

---

## Referee Comment (RC2) · Anonymous Referee #2 · 8 Jul 2020

General Comments

This paper presents a simple study of the way the 525 nm to 1020 nm-extinction coefficient ratio, a proxy for the particle size, evolves in function of the 1020 nm-extinction coefficient during the early period following the volcanic eruption. This study is based on ten test cases of volcanic eruptions that affected the stratosphere and were measured by eitther SAGE II, or SAGE III on ISS. The author's aim is to show that the joined evolution of these two parameters follows in many cases a simple relationship, regardless of the particle size distribution or the details of the circumstances of the

eruptions.

This study definitely lacks precision in my opinion. The whole analysis is based on approximations, quite subjective considerations and fast conclusions that remove convincing ground to the argumentation, yet based on a necessary limited sample of eruptions. Furthermore, since one of the two key dataset (i.e. SAGE III/ISS) shows a bias at one of the considered wavelengths, they interpolate these values using a simple law, while the situation after a major eruption like the ones studied here is expected to be complex. Building on approximate data with coarse approximations and arguments gives results that are sometimes questionable, especially in view of some applications the authors have in mind.

Concerning the text, the authors' formulation is in some cases so general that they make the described concept no more relevant, e.g.: "the stratospheric aerosol optical depth" [at 525 nm or 1020 nm], mentioned as if it was a fixed number. The authors should also be more accurate in their naming of the quantities they consider (e.g.: "[aerosols] extinction coefficient perturbation ratio", "perturbation aerosol extinction coefficient ratio", "perturbation extinction ratio", "perturbation ratio"). Overall, sloppiness brings a lot of confusion in the text and undermines the argumentation.

Concerning the case of the Mt Pinatubo, it is particularly surprising that this case is considered and discussed without any mention of the fact that SAGE II could not appropriately measure the extinction coefficient during several months after the eruption due to the extreme opacity of the volcanic cloud. This is however a major drawback for the present study.

Consequently, using this analysis and the relationship inferred by the authors to validate complex models as suggested by the authors seems far premature and overrated.

Specific comments

L. 15, p. 1: The integral over its whole domain of definition, of a distribution function

is equal to 1. The change in multiple orders of magnitude should concern the aerosol size number density, not the aerosol size distribution.

L. 24-25, p. 1: For such general consideration, I would suggest citing Robock, Rev. Geophys., doi: 10.1029/1998RG000054, 2000.

L. 1-2, p. 2: To illustrate the efforts to model the climate impact, I suggest citing some work by Timmreck et al., for instance, Timmreck et al., Geophys. Res. Lett, doi: 10.1002/2015GL067431, 2016.

L. 55-58, p. 2: This sentence is correct, but gives still a biased view of the reality: the SAGE II mission was exceptional in several senses, one of them being that it spanned a period with a particularly high amount of very large volcanic eruption. If it had been launched 10 years later, the situation would have been very different. The authors should be attentive to give a correct view of the reality.

L. 56, p. 2; L. 79, p. 3; L. 360, p. 13; L. 437-438, p. 26; caption Figure 1, p. 15: Kovilakam et al. is not published so far, and mentioning it as if it was the case is not ethical and should not be done. If this paper is not accepted in due time, please refer to another paper, e.g. Thomason et al., 2018.

L. 63, p. 2-L. 65, p. 3; L. 72-75, p. 3: This is not true and should be corrected. The ESA Envisat mission provided three experiments with high interest for aerosol studies: the SCIAMACHY spectrometer measuring in the UV-visible-near-IR range (von Savigny et al., doi:10.5194/amt-8-5223-2015, 2015; Noel et al., Atmos. Meas. Tech, doi:10.5194/amt-2020-113, in review, 2020), the IR limb sounder MIPAS (Griessbach et al., Atmos. Meas. Tech., doi:10.5194/amt-9-4399-2016, 2016), and the UV-visible-near IR stellar occultation instrument GOMOS (Bingen et al., Remote Sensing Env., doi: 10.1016/j.rse.2017.06.002, 2017). Also, the ACE-MAESTRO mission provided aerosol extinction from solar occultation measurements (McElroy et al., Appl. Opt., doi10.1364/AO.46.004341, 2007).

L.103-104, p. 4: In the case of the major eruptions that the authors consider, the situation is definitely more complicated than in this simple monodispersed aerosol model.

L. 106-108, p. 4, Figure 3: "The stratospheric aerosol optical depth" is vague. Please specify. Figure 3: SAGE II was not able to measure correctly the aerosol extinction during the months following the Mt Pinatubo due to the saturation of the atmosphere in aerosols. How do the authors infer the "stratospheric aerosol optical depth" during this period?

L. 125-126, p. 4: The authors should be specific: Which value of the 525 nm and 1020 nm aerosol extinction coefficient do they choose to infer "the impact of these eruptions"? The conversion of sulfur gases to sulfuric acid is a process that requires several weeks, and the presence of ashes may significantly influence the aerosol population in the early phase after the eruption, as mentioned before by the authors.

L. 128-129, p. 5: This is a serious setback for this study! A "simple Angstrom coefficient" interpolation states that the aerosol population has a simple structure, which is probably not the case in the post-eruption period where different aerosol modes (thin particles, ashes, aged aerosols) coexist. The interval [448 nm, 756 nm] is quite large and inferring the extinctin coefficient value at 521 nm is uncertain. Furthermore, while implementing an interpolation, why don't the authors interpolate at 525 nm, the wavelength effectively used in the paper?

L. 132-133, p. 5: How do the authors assess the difference between the interpolated value and a good approximate of the true value, if the SAGE III/ISS are biased?

Figure 4: Please indicate the Julian day corresponding to 1 January 1985. Is it Day 200?

L. 148-149, p. 5: Without indication of which points correspond to volcanically pertubed observations, and which ones to unperturbed periods, it is not possible to assess the pertinence of this sentence.

L. 149-150, p. 5: What is a "low extinction coefficient ratio"?

L. 150, p. 5: Was this hypothesis checked in some way?

L. 152-155, p. 5: If the authors clearly excluded any considerations about fire and pyrocumulus events, what do they mention them here? Or conversely, why did they reject such events in L. 69-70, p.3, if they use them here?

L. 158, p. 5: Since the time between two visits is about a few weeks to months, the observed evolution potentially starts at very different stages of plume development, possibly characterized by very different aerosol population features (importance of the ash fraction, development of the aerosol microphysics, mono/multimodal character of the size distribution, etc.).

L. 160, p.5: How do the authors select "the maximum value"? As shown in Figure 4, such situation is rich in outliers. Do the authors consider the highest outlier? Which is the relevance of such a choice, in a situation where the instrument measures locally, at a single time occuring after "a few weeks to months", more or less close to the eruption?

L. 161, p. 5 and Figure 4: What do the authors mean by a "9-point window"? Is each point in Figure 4 such a "maximum" in a 9-point window?

L. 162, p.6: "main aerosol layer" vs "entire layer": do the authors refer to the volcanic cloud, as opposed, say, to the combination of the Junge layer and the volcanic cloud?

L. 165, p. 6: What are the authors speaking about? "To produce a mean value" of what? "we required a minimum of 6 points": which kind of points and over which selection?

Figure 7: Same question as for Figure 3. Furthermore, the authors should add the Ruiz case to ease the comparison.

L. 172-178, p.6: This classification is poorly convincing. Concerning the events with "a

rapid increase in aerosol extinction coefficient and ratio following the eruption", this can-not characterize Ambae for which the increase in extinction ratio mainly spans about the 120 days preceding the eruption (as indicated by the anthors), saturating at the moment of the start of the eruption while the extinction coefficient just starts to increase. The authors find the cases of Ruang and Kelut as "following their own way" while these cases have many similarities with Raikoke that they see in another category. Clearly, the categorization is subject to interpretations, what makes the classification over an 8-eruption sample poorly convincing.

Figure 8a: The authors should show in some way, for each eruption, where is the "before" and where is the "after".

L. 179-181, p. 6: How do the authors define "the first data point"? Is it the extinction ratio value corresponding to the earliest time plotted in Figure 7? In Figure 7, the time interval considered between this first data point and the eruption start time is different in each case and the choice of this time interval obviously weights on the initial value of the extinction coefficient ratio, and hence of the "extinction coefficient perturbation ratio" shown in Figure 8b. Furthermore, in at least 2 of the 8 considered cases (Cerro Hudson and Manam), there is no clear indication at all that the start of the curve corresponds to any "baseline", and in several other ones (Kelut, Pinatubo, Ruang, Ulawun, Raikoke), the presence of such "baseline" is very uncertain. Finally, the scales used for both the extinction coefficient and the extinction coefficient ratio are different and differently related for all cases, making the establishment of parallel behaviour uncertain. Hence, drawing any conclusion on these values, taking into account the high uncertainty on the value of the maximum extinction [ratio] (See comment on L. 160, p.5) looks particularly hazardous.

L. 184-186, p. 6: See previous remark.

L. 197, p. 6-L. 201, p. 7: Beyond the uncertainties related to the values plotted in Figure 8b (See comment on L. 179-181, p. 6), this model is so coarse, ignoring the

fact that some eruption produce more sulfur gases and some other ones more ashes, ignoring all effects related to the geolocation and consequent dynamical features of the plume dispersion as well as seasonal effects, and ignoring intrumental limitations (notably SAGE II's "blindness" in the early phase of the Pinatubo eruption), that it seems absolutely premature to draw any conclusions from these 10 cases among which only a handful of the considered ones follow a nice flat curve. Applying such relationship to other data sets from limb sounders where retrieved extinction values depend yet on a set of assumption, is at risk to get quite far removed from the truth. Instead of suggesting using this tool for the evaluation of interactive aerosol modules for GCMs and ESMs, it would be useful, on the contrary, to use GCMs with elaborate aerosol microphysical models to assess if this relationship is sufficiently grounded to be used elsewhere.

L. 206-207, p. 7: If two closely related eruptions from a same volcano, occurring within such a short period of time, have yet to be splitted in a "regular case" and an "outlier", it becomes very hard to find any foundation in the proposed relationship. Even if other observations could bring some evidence that the plume composition, due to this quick succession, was very different in both cases, such argument could not be invoked since this kind of consideration has been excluded in this study.

L. 210-213, p. 7: Choosing the "initial state" of the second eruption of Ambae (with a value of 4.9 for the extinction ratio) is a highly speculative exercise ignoring the fact that the microphysical evolution is highly perturbed by the addition of fresh material. In no way, the situation corresponding to this value of 4.9 for the extinction ratio can be considered as a "baseline". Furthermore, the given values are quite arbitrary: the value of 5 is never reached by the extinction ratio (the maximum is about 4.7), and the extinction ratio decreases far below 4.1, toward a value lower than 3.5 and that cannot be estimated from the limited curve shown in Figure 7. Hence, putting both Ambae eruptions again (and in contradiction to what is observed in L. 206-207, p. 7, see comment) in the category of "regular cases" is particularly dubious.

L. 220-221, p. 7: This looks like a fast conclusion about a situation where microphysics, dynamics, and multiple injections of volcanic matter combine in a complex way, as illustrated in Figure 9.

L. 223, p.7: "the main aerosol layer", "all parts of the volcanic cloud": please clarify (See also comment on L. 162, p.6).

L. 226-228, p. 7: What do the authors mean by "the first observations"? Also, the end value is rather 2.9 than 3.

L. 228-229, p.7: Where do the authors find this value of 2.7? Such value is never reached in Figure 7, nor is indicated in Figure 8. Is the "perturbation extinction ratio" another quantity than the ones illustrated in these figures?

L. 230, p. 8: What is a "compact extinction coefficient [ratio]"?

L. 230-235, p. 8: It is surprising to give such an importance to outliers that in other circumnstances would be fastly overlooked.

L. 239, p. 8: What do the authors mean by "the spread of the Nevado del Ruiz, Cerro Hudson and Raikoke eruptions"?

L. 243-251, p. 8: This part of the text looks like a suite of speculations without attempt to analyse them seriously, making this enumeration not very useful.

L. 257-259, p. 8: The sentence is unclear. Please rephrase.

L. 263-266, p. 9: The sentence is unclear. Please rephrase.

L. 286, p. 9: What do the authors mean by "identical perturbations"?

L. 285-293, p. 9 and Figure 11: Basically, the extinction coefficient at 525 nm and 1020 nm and hence they ratio, can be exactly calculated by Mie theory for fixed values of the aerosol volume density and for single-radius particles. Hence, the differences found in Figure 11 for Nevado del Ruiz and Ambae only shows in which extend this simpel

model of single-radius particles departs from the reality. Furthermore, the authors mention coagulation effects, but ignore sedimentation which is nevertheless crucial after a large eruption, and high from the early post-eruption period. L. 289, p. 9: There is no single timescale used in Figure 11, nor in Figure 2, and the authors should thus not use this concept alone.

L. 305-307, p. 10: From the simplicity of this model and the fact that important aspects are neglected (e.g. the absence of sedimention which critically influences the extinction and extinction ratio), I do not think that any conclusion can be drawn about inferring primary microphysical effects from these SAGE II and III/ISS observations.

L. 323, p. 10: Which "measurement paradigms" are the authors talking about?

L. 327-328, p. 11: Using a tool to assess the data quality of a data set is only meaningful if this tool is not based on coarser assumptions and approximations as the ones leading to this data set. I am not sure that this is the case in the present work.

L. 331-333, p.11: Among the different applications proposed by the authors, the use of an extinction ratio value inferred from a relation established from the points 6-5-8-7-0-2 in Figure 8b to fix the size parameter used in the OSIRIS retrieval (e.g. through the Angstrom coefficient) seems to be the only one for which this simple model could have a real added value, in my opinion. It is really a pity if this model even cannot be beneficial in this framework.

Technical corrections

Caption Figure 9: It would be useful to indicate here the latitude of the Ambae volcano.

L. 216-217, p. 7 and caption Figure 9: the indications of wavelength and time are not consistent between the text and the caption. Please specify the exact time duration (start and end time).

---

## Referee Comment (RC3) · Anonymous Referee #3 · 20 Jul 2020

General comments

The paper describes the relationship between the perturbation in the aerosol extinction coefficient and the extinction ratio or particle size in the early months following small to midsize volcanic eruptions. The authors acknowledge the limitation of the analysis, which were restricted to early months following volcanic eruptions while tracking the main layer, mainly because of the presence of significant amount of ash. I believe that the analysis presented here are incomplete and the paper would benefit from expanding the analysis to a longer period (not only the early months) and wider range

of altitudes, rather than the peak of the aerosol layer. In addition, the paper needs clear and defined objectives (see my comment below).

Specific comments

Page 3, L64-66: 'space-based missions are mostly limited to single wavelength measurements associated with instruments such as . . .' The statement is only valid for the datasets used in GloSSAC climatology. CALIPO routinely provides aerosol measurements at two wavelength 532 and 1064 nm, while OSIRIS can provide measurements at 750 and 1530 nm, which were used to derive stratospheric aerosol particle size information (Rieger et al., 2014). In addition, MAESTRO (McElroy et al., 2007), GOMOS (Vanhellemont, et al., 2016), and SCIAMACHY (Taha et al., 2010; Malinina et al., 2019) can also provide aerosol measurements at multiple wavelengths. While the quality of the measurements is debatable, its existence is not.

Page 3, L73-74: 'It is also, until the start of the SAGE III mission, a period where the long-term stratospheric record is less robust due to the lack of global multiwavelength measurements of aerosol extinction coefficient.' Again, the Authors are describing the GloSSAC dataset rather than the global stratospheric records. See previous point.

Page 3, L76-81: 'Thus, the original aim of this work was to understand how volcanic events manifest themselves in SAGE II/III observations with the goal of 1) inferring the uncertainty in single wavelength space-based data sets that use a fixed aerosol size distribution as a part of their retrieval algorithm such as the OSIRIS and CALIOP and 2) infer how well the wavelength dependence can be estimated for these single wavelength measurements.' The authors failed to address both objectives and I don't see how the paper's findings, in its current form, can be of any use to these instruments because of the limited analysis shown her. I suggest either expanding the scope of the work to address those objectives or revising it to more realistic objectives.

Table 1: 'Volcanic eruptions and smoke events that significantly impact stratospheric aerosol levels in the Version 2.0 of the GloSSAC data set' Figure 1 shows very low

[Figure]

aerosol and no impact of any of the volcanoes listed between 1998-2004, which implies that none of these eruptions reached the stratosphere. I suggest revising Table 1 by removing all volcanic eruptions that is not seen by GloSSAC. In addition, add the eruption altitude, similar to Table 2.

Figure 3: Unlike the rest of the analysis shown in this paper, the figure is for GloSSAC zonal mean aerosol stratospheric optical depth and ratio rather than extinction coefficient and ratio for the peak aerosol layer. I suspect that the extinction ratio is inaccurate, given that SAGE II measurements were missing following the early months of the eruption, and the dataset was mostly reliant on single wavelength Lidar measurements. For the sake of consistency, I suggest using SAGE II measurements of the aerosol extinction and extinction ratio, similar to Figures 4-7.

Figure 4a: Can you use different color for the extinction values at 20.5 km?

Figure 4b: Can you plot all data shown in Figure 4a while using different color for extinction ratio at 20.5 km?

Figures 4, 6, and 7: The Days label is confusing. Can you use month/year or something similar?

Figure 5: change 'Same data as shown in Figure 3' to Same data as shown in Figure 4a or 4b. In addition, can you specify if the data shown are for 20.5 km or all altitudes? If so, can you use different color for 20.5 km.

Page 6, L189 'the maximum extinction coefficient at 525 nm does not necessarily occur at the same altitude or time as the maximum in 1020 nm extinction coefficient'. This is all the more reason to track the plume at different altitudes/zones rather than the maximum extinction value.

Figure 7: I suggest over plotting points symbol to show the number of points used in each figure, adding a second vertical line for Ambae and Ulawun denoting the second eruption date, and using the same x-axis scale for all figures.

Figure 9 and page 7, L220 'It is clear here that the maximum in the extinction ratio lies below the main peak in extinction coefficient in the tropics and, notably stretches to higher southern latitudes and the maximum values actually occurs near 30° S despite more inhomogeneous conditions at this latitude than in the tropics' and page 8 L240: 'Both eruptions show increased aerosol extinction coefficient ratios away from the main aerosol peak suggesting, at least in part, behavior more consistent with most eruptions.' This is interesting observation that raises more concerns about the analysis shown in Figures 6 and 7. Is it possible that tracking the maximum extinction value is not the best approach as it might bias the outcome, especially where the aerosol extinction is very large? Very large extinction values can be caused by the presence of Ash particles, or it is an artifact in SAGE measurement when the volcanic plume is localized and spatially inhomogeneous. In addition, the result results shown here can be easily biased by SAGE limited coverage. Perhaps repeating Figure 7 using zonal means at different altitudes and extending period of the analysis can produce a more consistent relationship between the aerosol extinction and extinction ratio perturbations.

Page 8, L246: 'peak extinction level as essentially all the data follows the mean relationship in Figure 7g.' The sentence is unclear. This correct for Ulawn, however, the paragraph is discussing Raikoke eruption. Please revise the sentence.

Page 8, lin3 253: 'It is also possible that a pyrocumulus event, that occurred in Alberta, Canada just prior to the Raikoke eruption, plays a role in the evolution of extinction following this event.'. In addition to Alberta, there was a second PyroCb event in Siberia, Russia in July 2019, that reached the stratosphere and was also seen by SAGE III/ISS (https://directory.eoportal.org/web/eoportal/satellite-missions/i/iss-sage-3, Figure 7). It is more likely that the smoke aerosol interfered with the Raikoke analysis. If the two different aerosol layers were separated (which is most likely), then repeating the analysis at different altitudes instead of tracking the peak extinction can explain the behavior seen in Figure 7h.

Section 4: The first paragraph leading to the aerosol perturbation model relies on unsupported speculations (by the authors own admission), and it can benefit from the addition of few references that support those assumptions.

I also find the whole discussion regarding the model calculation and figure 11 confusing and I can't relate the figure to what was presented in the previous section. The aerosol extinction ratio perturbations for Ambae are almost double those for del Ruiz eruption and figure 11b in particular shows large extinction ratios (>5) not seen by any of the cases shown in this paper. Close inspection of both figures indicate that the extinction ratio perturbation is very sensitive to the baseline ratio, and if Manam or Ulawn baseline values were used, the extinction ratio values would've been even higher.
* * *

---

## Author Comment (AC1) · 15 Oct 2020

**Pasquale Sellitto (Referee)** pasquale.sellitto@lisa.u-pec.fr

The manuscript ''Evidence for the predictability of changes in the stratospheric aerosol size following volcanic eruptions of diverse magnitudes using space-based instru- ments, Thomason et al.'' discusses the relationship of multiwavelength aerosol extinc- tion observations, in the first phases of volcanic plumes dispersion following small-to- strong stratospheric eruption, and the apparent size of the particles in the plume. From my perspective, the main result of this work is the evidence of a clear relationship of the extinction ratio (shorter-to-longer wavelength, namely 525/1020 nm, hereafter referred to as ER) and the strength of the eruption in terms of the aerosol extinction at 1020 nm (hereafter referred to as AE), see Fig. 8b. This relationship is associated to the apparent particles size in the aerosol layer (Fig. 2). Even if this behaviour is probably expected, I personally think that a systematic study of this relationship is very interest- ing and important. The identification of volcanic plumes by means of the concurrent increases of the aerosol extinction (or the integrated optical depth) and modifications of the spectral variability of this extinction (or the Ångström exponent) has been exploited, in the past, in different studies – and it is a tool that I personally use a lot. Nevertheless, a systematic effort to: a) study this aspect over a long observation series, or even construct a theoretical basis, has never been attempted, to my knowledge. Thus, I think that such kind of work would deserve immediate attention and rapid publication. Unfortunately, the present manuscript, while having all elements to provide the com- munity with both points a and b mentioned above, is somewhat flawed in two aspects, that I mention in the following. I encourage the Authors to tackle these two "major" issues, as well as a number of specific issues that I also list in the following, and I'll be happy to re-evaluate the manuscript revised accordingly, once the due modifications are done.

Sincerely, Pasquale Sellitto

*Major comments:*
1) The conceptual model defined to connect the apparent radius of the aerosol layer and AE/ER, as defined at L271-273 and shown in Figs. 2 and 11, is not completely clear to me. The model is defined for what looks like a monodispersed aerosol layer (which is also mentioned at L102), which is fine when discussing this in theory (Fig. 2) but is a little bit odd when applying to real data (Fig. 11). In the real world, the size distribution would not be monodispersed, so I guess a more realistic size distribution should be used, which is still not very complicated using a mono-modal size distribution with varying mean radii. Another thing that gets me confused is that in the equation at L271, it looks to me that the numerator of the ratio on the right is not the perturbation delta_k but rather k (otherwise, you don't have number density n(r) but the perturbation in number density delta_n(r)). Like this, it looks like you're accounting twice for the background, when you later scale the result with respect to the background. And also, the Mie scattering efficiency is calculated with a Mie code (which one?) and is based on an assumption of the particles' composition (their refractive index): which is your assumption? If, as I imagine, the Authors have supposed a pure sulphates plume, why the ash has been neglected – it may be important in the first phases of some eruptions, e.g. Raikoke and Kelud? And again, if it is sulphates, what has been supposed for the mixing ratio of sulphuric acid in the sulphate aerosols droplets, which is a factor that can modulate the extinction of the particles, at least at longer wavelengths? By using the Mie scattering coefficient, you suppose that the absorption can be completely neglected. If ash is to be considered, this might not be completely true. All these aspects have to be clarified or maybe the model has to be slightly refined to account for the mentioned problems.

*We did not clearly explain the purpose of the model discussed in section 4. We are not suggesting that the simple model used there is completely realistic as it only accounts for sulfuric acid aerosol and models the perturbation as a single particle size rather than a distribution of some kind. For the latter limitation, we have limited or no information on the changes to the aerosol size distribution for these events at the latitudes and times of the space-based observations. The model is an attempt to justify our interpretation of the observations shown in figures 7 and 8 but not to exactly model any event.*

*The n(r) parameter is for the perturbation and this has now been noted in the text. We are indeed modelling only sulfate aerosol for which there is no absorption at the wavelengths consider in this paper (and scattering and extinction cross sections are the same). We updated the text to indicate that we are, in fact, using the extinction cross sections and the Mie code is based on Bohren and Huffman (1998).*

The manuscript structure has to be improved. In the present version, the main re- sults and the overarching narrative are not completely clear. First, the Introduction fails to present the motivations of this work. Some elements of motivation are in Sect. 2 rather than in the Introduction. It is stated two times that "The primary goal of this effort was to assess data quality of data sets consisting of a single wavelength mea- surement of aerosol extinction coefficient or similar parameter particularly when a fixed aerosol size distribution is a part of the retrieval process." but I cannot see where this is discussed in the text. As stated in my introduction, I would rather say that the main motivation for this study (and it is an important motivation!) is to develop a means to identify volcanic plumes and to classify them based on the eruption strength, and I suggest mentioning this as a motivation. Section 2 does not present satisfactorily the SAGEII-III datasets and a lot of information is lacking. Care should be taken, through- out the manuscript, to

introduce the Figures sequentially in the text and not to discuss them before presenting their content.

*We have improved the introduction particularly in providing better motivation for the study contained in the remainder of the paper. Figures are now discussed in order of their number.*

I add some specific points in the following Specific Comments.
*Specific comments:*
1) I don't understand the sense of the word "predictability" in the title, as you attempt no predictions

*Clarified later in the text as referring to the ability to predict the impact on OSIRIS observations in the absence of SAGE-like observations.*

2) L15-16: there is a lacking mention to the ER, which is the measured parameter that actually is studied and vary for different eruptions strengths (and AE)

*Added*

3) L16: "The relationship is measurement-based and does not rely on assump- tions about the aerosol size distribution.": strictly speaking there is an assumption of monodispersed aerosol layer

*The primary results are those found in Figure 8 showing how the aerosol extinction coefficient perturbation ratio varies with aerosol extinction coefficient. These results are entirely measurement-based. The model is only used to explain (as is clarified in the text) to explain why we believe what we observe in the measurements (at least for the 8 events that follow the main curve) is consistent with homogeneously nucleation of many small aerosol.*

4) L18-22: "Despite this limitation...particle size": these two points (the use in eval- uating global models and the improvement of mono-spectral AE observations) is not discussed in depth in the manuscript so it is strange that it is mentioned in the Abstract

*We've improved this aspect of the discussion in section 4.*

5) As stated at Major Comment #2, I feel that the Introduction is very synthetic and does not make a good job in motivating this work

*The introduction has been expanded and better reflects the goals of this paper.*

6) L24: "Eruptions of volcanoes" –> "Volcanic eruptions"

*Done*

7)  L25: "Volcanically-derived aerosol": Here the you talk in general of volcanically derived aerosol and, later in the text, the discussion specialises on sulphates. A line is probably lacking here on the mention of the possible variety of volcanic particles (ash and sulphates)

*Done*

8)  L29: I think that Tang et al. 2013 do not discuss of the impact of volcanic aerosol on transport but of the decrease of ozone in the stratosphere and then, consequently, in the troposphere by stratosphere-troposphere exchanges. As one lines above you discuss about the radiative heating of volcanic aerosols, one can erroneously think that Tang et al. talk about radiative-dynamical interactions and the crossing of tropopause by plume self-lifting, which is not the case. Please correct

*Changed the reference to:*
*Pitari, G., Cionni, I., Di Genova, G., Visioni, D., Gandolfi, I., and Mancini, E.: Impact of Stratospheric Volcanic Aerosols on Age-of-Air and Transport of Long-Lived Species, Atmosphere-Basel, 7, 149, ARTN 149*

9)  L31: ". . .either the measurements. . .": please develop a bit to clarify how measure- ments have been used in global models to derive climate impact of volcanic eruptions

*Clarified this statement*

10)  Why not more classic section titles: "2. Data and Methods" and "3. Results"?

*Fine.*

11)  L45: "well-known SAGEII": it is well-known but it might be less known for a part of the readership of ACP. Thus, please suppress "well-known" and mention the years and months of operations of SAGEII, from launch to end of mission.

*Removed.*

12)  In Tab. 1 there are more eruptions than what studied in Sect. 3 and 4 and there are fires as well. While later in the text it is said that fires are not in the scopes of this work (while it would have been interesting to see where the points in Fig 8b locate for Canadian and Australian fires. . .), it is not clear to me why many eruptions possibly present in the datasets have not been included in this study: Sarychev, Kasatochi and Nabro are largely considered "major moderate eruptions" for their impact on the stratosphere but are neglected in this study.

*The fire events tend to align more with Raikoke than with what we interpret as sulfuric acid dominant eruptions. We felt including them distracted from the main goals of the paper since there is little reason to expect them to behave the same as a volcanic eruption. Some events in Table 1 occur during periods where SAGE measurements do not exist and thus are not included in the analysis.*

13) L55: Please briefly introduce this GloSSAC dataset. Also please mention in the corresponding reference that the relative manuscript is presently under discussion/review and add a link for the discussion paper

*Added a bit of material here. The paper has been accepted for publication. The reference was updated to the discussion but we plan to update to the final reference before this paper is complete.*

14) L59: "SAGEII record": here is very clear that mentioning the start-end of SAGEII operations is important

*Updated to include month..*

15) L61: "…subtly modulate climate…": probably it can be mentioned that the ag- gregated impact of these "small-to-moderate" eruptions is significant (see also Ri- dley, D. A., et al. (2014), Total volcanic stratospheric aerosol optical depths and implications for global climate change, Geophys. Res. Lett., 41, 7763– 7769, doi:10.1002/2014GL061541)

*Added*

16) L63: as for SAGEII, the precise period of operations of SAGEIII/ISS should be clearly written

*The mission is on-going so data is available from June 2017 through the present.*

17) Fig. 1: the periods of operations (start/end of missions) of SAGEII and III should be probably indicated in this figure, e.g. with vertical dotted lines. That would help a lot in the understanding of the different discussions of Sect. 2

*Added*

18) L67: "Raikoke…2020): why not arranging these eruptions in chronological order?

*Reordered.*

19) L69: in the following paper about Canadian fires 2017, a similar method as the one discussed in this manuscript is used to

identify and separate a fire plume from an anthropogenic plume (see their Fig. 2b): Kloss et al. Transport of the 2017 Canadian wildfire plume to the tropics via the Asian monsoon circulation, Atmos. Chem. Phys., 19, 13547–13567, https://doi.org/10.5194/acp-19-13547-2019, 2019.

*Added.*

20)  L69: please note this pre-print manuscript on the Australian fires 2019-20: *https://arxiv.org/abs/2006.07284*

*To our understanding, this is not considerable referenceable by Copernicus. However, we have added a similar reference.*

21)  L71-72: ". . .a qualitative difference. . .": please mention a difference with respect to what? (That would be clearer if the periods of operations of SAGEII and III are indicated in Fig. 1)

*We've rewritten this section to make the qualitative difference clearer.*

22)  L73-80: These motivations should be moved to the Introduction

*This discussion is now in the introduction*

23)  L81-93: also more appropriate in the Introduction?

*We think this discussion needs to be located here since it deals with issues related to the instrument and the analysis.*

24)  L95: what do you mean with "robust"?

*Updated to 'high accuracy and precision'*

25)  Are footnotes allowed in ACP format?

*Removed*

26)  L101-102: see Major Comment #2

*Mie calculations are based on the code provided in Bohren and Huffman (1998). We've also changed the text to indicate that we are discussing single particles (at this stage) not a size distribution.*

27) L124-125: "These begin...Table 2)": Is this line to be moved earlier (e.g. L120)?

*Moved.*

28) L129: "...O4 absorption...version.": Is there a reference for this underestimation? And also, a few words should be included to clarify why using the interpolation between 448 and 756 nm limits (or avoids) this underestimation: O4 has no absorption at these wavelengths and has a significant absorption at 521 nm?

*The O4 error has a subtle (positive) impact on the ozone retrieval below 20 km where there is significant overlap in the spectral regions used to retrieve ozone and where O4 absorbs. The small error in ozone has a larger impact on aerosol where ozone absorbs strongly (521, 602 and 676 nm) but other aerosol measurement wavelengths are unaffected.*

29) L131: "...(602 nm... 521 nm)": why? (see previous comment)

*See above*

30) L138: "...November 13...": please mention the year

*Added*

31) 142: "The opposite of what..." → "This is the opposite of what..."

*Updated*

32) L143: "The extinction ratio becomes...": here talking about Nevado del Ruiz (not Pinatubo)?

*Ruiz, clarified.*

33) Why not aggregating Fig 4 and Fig 5 in a unique figure with 3 panels?

*We preferred it this way.*

34) Fig. 4: Why not restricting the yaxis scale to something like 2 to 4? There are no values <2 or >4.

*Done*

35)  Fig. 4 caption: What do you mean with "the scatter"?  I would just say "the time series of 1020... and 525 to 1020..."

*Changed*

36)  Fig. 5: why not evidencing the pre-eruptive points (the cluster for smaller values of the 1020-nm AE) with a different colour or different symbol? Can it be possible to identify the points in the earlier stage of eruption, as well (so to corroborate the hypothesis of a different cluster of values, i.e. with ash)?

*Done*

37)  L148-149: "The distinction...recognizable": this can put more in evidence in Fig. 5 (see previous comment

*Figure enhanced as requested.*

38)  L150: Has a sulphuric acid hypothesis been proposed earlier?  A precise point where ash is discarded is not present in the previous text

*This is now introduced more clearly in the introduction*

39)  L151-152: "Generally,...events": Can this be shown more clearly?

*We've clarified this discussion along with the discussion in comment 40.*

40)  L153-156: "This was particularly...higher latitudes events": all this part is not very clear to me

*See above.*

41)  L193-194: "At some point...reasonable": this makes reference to Fig. 2?  Please clarify

*Added reference to Figure 2.*

42)  L197-201: "This relationship. . .space-based instruments": how this can be done (inferring uncertainty in mono-spectral observations and evaluating aerosol modules in GCMs)? I feel that this should be discussed much more in depth

*We have clarified this discussion to point out that since models with detailed aerosol microphysical models possess knowledge of the composition and size distribution of aerosol in space, it is a straightforward calculation to produce extinction at any wavelength.*

43) L202-: in general, in Section 3 and 4 discussions of volcanic events considering existing information in the literature are lacking. For Ambae, for example, in the paper already cited, Kloss et al., 2020 (by the way , please correct the reference as this paper is now published in JGR and no more in preprint), the plumes are detected using SAGEIII observations and simultaneous increases of the AE and the partial column Ångström exponent (Fig. 8 of Kloss et al.). This can be easily put in connection with Fig. 7f.

*Added.*

44) L226: "…Ruang…": please add year of eruption

*Done*

45) L230-232: "The Kelud. . .um)": here is an example where your work can be put in context with existing literature. In the following paper, it has been shown that ash was present for a long time in Kelud plume: Vernier, J.-P., Fairlie, T. D., Deshler, T., Natarajan, M., Knepp, T., Foster, K., Wienhold, F. G., Bedka, K. M., Thomason, L., and Trepte, C. (2016), In situ and space-based observations of the Kelud volcanic plume: The persistence of ash in the lower stratosphere, J. Geophys. Res. Atmos., 121, 11,104– 11,118, doi:10.1002/2016JD025344.

*Sorry this paper is on the 2014 eruption of Kelud whereas we are discussing the 1990 eruption for which no comparable data exists.*

46) Fig. 7: it would be useful to have the indications (red dashed lines) for the two eruptions of Ambae and Ulawun

*Done*

47) Fig. 7 caption: the mention to the volcanoes names is probably redundant, as the volcanoes are also mentioned in the panels. Also "…vertical dashed lines." –> "…vertical red dashed lines."

*Done*

48) Fig. 8b: Why not quantifying this trend (linear regression and correlation parame- ter)?

*We thought about this but decided to not do so at this time because there are sufficient questions in our mind regarding the details of this relationship (particularly linearity in log-extinction coefficient/extinction ratio space).*

49) L242: ". . .(possible ash). . .": but also sulphate-coated ash or large sulphates in the accumulation mode possible, how do you

exclude these?

*Also possibilities as well as perhaps some ice crystals. Pure sulfuric acid droplets seem unlikely unless they are directly injected as droplets but with the lack of composition information, ash is a surmise. Clarified.*

50)   L248-250: "For instance, for Raikoke...and ratio.":  this makes reference to the  (complicated) issue of the mixing state of the aerosol population, including the possibility that ash is sulphate-coated and/or these particles may freeze. Probably a discussion  about that is needed here.
*We have included a brief discussion of alternative compositions*

51)   L250-251: "It is also possible...this event.":  interesting,  can you please develop this point a bit?

*There was a pyroCB about a month before the Raikoke eruption that SAGE II observed at about 12 km. After the eruption the two events became indistinguishable rather rapidly so how they interacted, if at all, is an interesting topic for consideration. For instance, SAGE II and CALIPSO observed a blob of aerosol that originated at high latitudes slowly rose through the stratosphere and ends up near 25N. This behavior is not typical of volcanic material (to our experience) but is fairly common for smoke events like a pyroCB (reference to this blob is included). Whether the smoke material managed to pass through the volcanic layer in some way or if the blob is some sort of mixed aerosol material including sulfuric acid coated smoke particles, is difficult to assess at this time. At this point, we know that it is possible that they two events interacted and this is the subject of current research and a forth coming publication.*

52)   L257: "These are initially. . .Figure 11)": How is it visible in Fig.  11. And also, why  Fig.  11 is discussed before its content is defined (in the following lines)?

*Updated to reference Figure 2a.*

L257: "...but coagulate...": if talking about coagulation, why not of heterogeneous  nucleation/condensation over pre-existing particles (sulphate aerosol or ash)? At this  point, it  looks clear to me that a discussion on the mixing state and aerosol micro- physics is quite needed

*In this case, we are discussing what we infer to be a process that produces the observations we report.  Condensation onto existing particles or the rapid scavenging of these new small particles would always decrease the extinction ratio as particles would become systematically larger. We cannot exclude that this process happens at some level but that it is not consistent with the observations reported herein.*

53)   Equations at L270-273 and inherent discussion: see Major Comment #1

*This discussion has been clarified as indicated in the response to Major Comment #1*

54)   Please add equation numbering

*Added*

55)   The priority of argumentations in the Conclusions is not clear to me. The main results of this work are probably the evidence of the dependence of the ER from the eruption intensity and AE (Fig 8b) but this is not even mentioned in the Conclusions

*We have clarified the goals and outcomes from this study both in the introduction and in the conclusions including referencing the key findings shown in Figure 8b.*

56)   L327-328: "The primary goal...process": as mentioned above (Major Comment #2 and Minor Comment #4) this is not discussed in the text, so it is strange to see this in the Conclusions

*We have clarified the difference between a long term goal (or motivation) of improving OSIRIS-like observations and the goal of this paper which is to establish how small-to-moderate volcanic events manifest themselves in SAGE-like observations.*

57) L330-331: "It is clear. . .therein": this is actually not very clear to me: in the text there is no assumption on the aerosol chemical composition.

*We've clarified that the model used in Section 4 is based on sulfuric acid aerosol. Here we've added material that notes that the model is homogeneous nucleation of very small particles that subsequently coagulate.*

---

## Author Comment (AC2) · 15 Oct 2020

General Comments

This paper presents a simple study of the way the 525 nm to 1020 nm-extinction coef- ficient ratio, a proxy for the particle size, evolves in function of the 1020 nm-extinction coefficient during the early period following the volcanic eruption. This study is based on ten test cases of volcanic eruptions that affected the stratosphere and were mea- sured by eitther SAGE II, or SAGE III on ISS. The author's aim is to

show that the joined evolution of these two parameters follows in many cases a simple relationship, regardless of the particle size distribution or the details of the circumstances of the eruptions.

This study definitely lacks precision in my opinion. The whole analysis is based on approximations, quite subjective considerations and fast conclusions that remove con- vincing ground to the argumentation, yet based on a necessary limited sample of erup- tions. Furthermore, since one of the two key dataset (i.e. SAGE III/ISS) shows a bias at one of the considered wavelengths, they interpolate these values using a simple law, while the situation after a major eruption like the ones studied here is expected to be complex. Building on approximate data with coarse approximations and arguments gives results that are sometimes questionable, especially in view of some applications the authors have in mind.

Concerning the text, the authors' formulation is in some cases so general that they make the described concept no more relevant, e.g.: "the stratospheric aerosol optical depth" [at 525 nm or 1020 nm], mentioned as if it was a fixed number. The authors should also be more accurate in their naming of the quantities they consider (e.g.: "[aerosols] extinction coefficient perturbation ratio", "perturbation aerosol extinction co- efficient ratio", "perturbation extinction ratio", "perturbation ratio"). Overall, sloppiness brings a lot of confusion in the text and undermines the argumentation.

Concerning the case of the Mt Pinatubo, it is particularly surprising that this case is considered and discussed without any mention of the fact that SAGE II could not ap- propriately measure the extinction coefficient during several months after the eruption due to the extreme opacity of the volcanic cloud. This is however a major drawback for the present study.

Consequently, using this analysis and the relationship inferred by the authors to vali- date complex models as suggested by the authors seems far premature and overrated.

*These points (above) are dealt with as specific points below. We have corrected the usage of perturbation terminology. Since much of the finding we show below are based on observations and, in fact, could be inferred from SAGE II observations alone (whose mission ended in 2005), we argue that the basic observationally-based findings as shown in Figures 7,8, and 9 are overdue.*

Specific comments

L. 15, p. 1: The integral over its whole domain of definition, of a distribution function is equal to 1. The change in multiple orders of magnitude should concern the aerosol size number density, not the aerosol size distribution.

*Corrected to clarify that the 'several orders of magnitude' refers to extinction coefficient rather than size distribution*

L. 24-25, p. 1: For such general consideration, I would suggest citing Robock, Rev. Geophys., doi: 10.1029/1998RG000054, 2000.

*Done*

L. 1-2, p. 2: To illustrate the efforts to model the climate impact, I suggest citing some work by Timmreck et al., for instance, Timmreck et al., Geophys. Res. Lett, doi: 10.1002/2015GL067431, 2016.

*Done*

L. 55-58, p. 2: This sentence is correct, but gives still a biased view of the reality: the SAGE II mission was exceptional in several senses, one of them being that it spanned a period with a particularly high amount of very large volcanic eruption. If it had been launched 10 years later, the situation would have been very different. The authors should be attentive to give a correct view of the reality.

*We have clarified this statement.*

L. 56, p. 2; L. 79, p. 3; L. 360, p. 13; L. 437-438, p. 26; caption Figure 1, p. 15: Kovilakam et al. is not published so far, and mentioning it as if it was the case is not ethical and should not be done. If this paper is not accepted in due time, please refer to another paper, e.g. Thomason et al., 2018.

*This paper is in the final stages of the publication process. We have updated the reference to the Discussions paper but we should be able to update to the final reference before this paper is completed.*

L. 63, p. 2-L. 65, p. 3; L. 72-75, p. 3: This is not true and should be corrected. The ESA Envisat mission provided three experiments with high interest for aerosol stud- ies: the SCIAMACHY spectrometer measuring in the UV-visible-near-IR range (von Savigny et al., doi:10.5194/amt-8-5223-2015, 2015; Noel et al., Atmos. Meas. Tech, doi:10.5194/amt-2020-113, in review, 2020), the IR limb sounder MIPAS (Griessbach et al., Atmos. Meas. Tech., doi:10.5194/amt-9-4399-2016, 2016), and the UV-visible- near IR stellar occultation instrument GOMOS (Bingen et al., Remote Sensing Env., doi: 10.1016/j.rse.2017.06.002, 2017). Also, the ACE-MAESTRO mission provided aerosol extinction from solar occultation measurements (McElroy et al., Appl. Opt., doi10.1364/AO.46.004341, 2007).

*While in retrospect, this sentence could be deleted completely, we have included the references to the ENVISAT instruments.*

L.103-104, p. 4: In the case of the major eruptions that the authors consider, the situ- ation is definitely more complicated than in this simple monodispersed aerosol model.

*In this case, we have changed the reference to simple aerosol radius rather than reference size distribution.*

L. 106-108, p. 4, Figure 3: "The stratospheric aerosol optical depth" is vague. Please specify. Figure 3: SAGE II was not able to measure correctly the aerosol extinction during the months following the Mt Pinatubo due to the saturation of the atmosphere in aerosols. How do the authors infer the "stratospheric aerosol optical depth" during this period?

*We have changed the text to reference the GloSSAC v2.0 data set. Descriptions of how missing data are accounted for are explained in the Kovilakam paper and the v1.0 paper (Thomason et al., 2018) in detail.*

L. 125-126, p. 4: The authors should be specific: Which value of the 525 nm and 1020 nm aerosol extinction coefficient do they choose to infer "the impact of these eruptions"? The conversion of sulfur gases to sulfuric acid is a process that requires several weeks, and the presence of ashes may significantly influence the aerosol pop- ulation in the early phase after the eruption, as mentioned before by the authors.

*We have clarified this statement.*

L. 128-129, p. 5: This is a serious setback for this study! A "simple Angstrom coef- ficient" interpolation states that the aerosol population has a simple structure, which is probably not the case in the post-eruption period where different aerosol modes (thin particles, ashes, aged aerosols) coexist. The interval [448 nm, 756 nm] is quite large and inferring the extinctin coefficient value at 521 nm is uncertain. Furthermore, while implementing an interpolation, why don't the authors interpolate at 525 nm, the wavelength effectively used in the paper?

*We agree that the bias is certainly inconvenient for this study, however we do not believe that this interpolation introduces significant uncertainty into the analysis. The new text:*

*We have replaced the 521 nm data product with an interpolation between 448 and 756 nm that employs a simple Angstrom coefficient scheme (602 nm and 676 nm measurements have biases similar to like those at 521 nm). This is possible since the stratospheric aerosol extinction coefficient is always observed to be smoothly varying with wavelength and approximately linear in log-log space. The presence of the 521 nm bias is inferred using this methodology and this approach was used in the validation paper for SAGE III/Meteor 3M aerosol data (Thomason et al., 2010). The differences between the inferred 521 nm extinction coefficients and the reported values in the lower stratosphere (tropopause to 250 km) average about 6% and are usually less than 10%. Above 20 km the differences are usually on the order of 1 to 2% with the estimate usually less than the observation. This difference is probably a reflection of the limitation of the accuracy of the interpolation and consistent with past uses of the same approach (Thomason et al., 2010). In any case, the effects of small to moderate volcanic eruptions on aerosol extinction coefficient as a function of wavelength described below are consistent whether 448 or 521 nm aerosol extinction coefficient in used in the SAGE III analysis. We interpolate the 521 nm values solely for comparison purposes with SAGE II data and this process has minimal impact on the conclusion drawn below.*

L. 132-133, p. 5: How do the authors assess the difference between the interpolated value and a good approximate of the true value, if the SAGE III/ISS are biased?

*See above text.*

Figure 4: Please indicate the Julian day corresponding to 1 January 1985. Is it Day 200?

*This figure is in days since 1 January 1985 which would be day 1 which is now noted in the caption.*

L. 148-149, p. 5: Without indication of which points correspond to volcanically pertubed observations, and which ones to unperturbed periods, it is not possible to assess the pertinence of this sentence.

*We have updated the figure to color the pre-eruption observations red.*

L. 149-150, p. 5: What is a "low extinction coefficient ratio"?

*We have updated this text to refer to extinction coefficient ratios close to and occasionally less than observed prior to the eruption (<2.3 or so)*

L. 150, p. 5: Was this hypothesis checked in some way?

*SAGE-like observations contain little or no information on composition. We have updated the text to indicate that the composition is unknown.*

L. 152-155, p. 5: If the authors clearly excluded any considerations about fire and pyrocumulus events, what do they mention them here? Or conversely, why did they reject such events in L. 69-70, p.3, if they use them here?

*This is simply a comparison of low vs high latitude events and their differences in zonal variability. The nature of the events is, in this case, immaterial.*

L. 158, p. 5: Since the time between two visits is about a few weeks to months, the observed evolution potentially starts at very different stages of plume development, possibly characterized by very different aerosol population features (importance of the ash fraction, development of the aerosol microphysics, mono/multimodal character of the size distribution, etc.).

*We agree that this is a complicating factor as we discussed L81-93, p3*

L. 160, p.5: How do the authors select "the maximum value"? As shown in Figure 4, such situation is rich in outliers. Do the authors consider the highest outlier? Which is the relevance of such a choice, in a situation where the instrument measures locally, at a single time occuring after "a few weeks to months", more or less close to the eruption?

*The data is well behaved and low noise particularly when the aerosol extinction levels are enhanced by an eruption. This process is primarily a way to find the altitude (and the associated extinction coefficients) of the volcanic layer in each profile where it can vary from profile to profile within a temporal bin and over the months following the eruption. Other strategies were tried but none particularly changed the results. We have clarified the text.*

L. 161, p. 5 and Figure 4: What do the authors mean by a "9-point window"? Is each point in Figure 4 such a "maximum" in a 9-point window?

*Figure 4 is for 20.5 km. We have clarified the text that the window is effectively a 4 km window in each vertical profile (9 points in the 0.5 km vertical resolution profiles).*

L. 162, p.6: "main aerosol layer" vs "entire layer": do the authors refer to the volcanic cloud, as opposed, say, to the combination of the Junge layer and the volcanic cloud?

*In this case, it works out to basically the same thing but we are referring (and have clarified) that we focused on the volcanic cloud.*

L. 165, p. 6: What are the authors speaking about? "To produce a mean value" of what? "we required a minimum of 6 points": which kind of points and over which selection?

*We moved this discussion forward in this paragraph. Particularly in the tropics, there are occasions where the number profiles in a temporal bin is very low and average statistics and sampling is pretty poor. We found that requiring at least 6 profiles in a temporal event caught all of these poorly sampled periods. We have clarified this in the text.*

Figure 7: Same question as for Figure 3. Furthermore, the authors should add the Ruiz case to ease the comparison.

*Done.*

L. 172-178, p.6: This classification is poorly convincing. Concerning the events with "a rapid increase in aerosol extinction coefficient and ratio following the eruption", this can- not characterize Ambae for which the increase in extinction ratio mainly spans about the 120 days preceding the eruption (as indicated by the anthors), saturating at the mo- ment of the start of the eruption while the extinction coefficient just starts to increase. The authors find the cases of Ruang and Kelut as "following their own way" while these cases have many similarities with Raikoke that they see in another category. Clearly, the categorization is subject to interpretations, what makes the classification over an 8-eruption sample poorly convincing.

*The Ambae plot shows the effects of two eruptions occurring in April and July 2018. For the second event, the extinction ratio increases somewhat (on the scales shown in the plot) earlier than does 1020-nm aerosol extinction coefficient. We have simplified the categories to two types: one in which the aerosol extinction ratio increases (or remains large) relative to the baseline and ones where the initial change decreases the aerosol extinction ratio followed by some recovery to larger values.*

Figure 8a: The authors should show in some way, for each eruption, where is the "before" and where

is the "after".

*Since extinction coefficient always increases following an eruption, we have noted that the before points are always the left most data value is the before value and the right hand value is the after data.*

L. 179-181, p. 6: How do the authors define "the first data point"? Is it the extinction ratio value corresponding to the earliest time plotted in Figure 7? In Figure 7, the time interval considered between this first data point and the eruption start time is different in each case and the choice of this time interval obviously weights on the initial value of the extinction coefficient ratio, and hence of the "extinction coefficient perturbation ratio" shown in Figure 8b. Furthermore, in at least 2 of the 8 considered cases (Cerro Hudson and Manam), there is no clear indication at all that the start of the curve corresponds to any "baseline", and in several other ones (Kelut, Pinatubo, Ruang, Ulawun, Raikoke), the presence of such "baseline" is very uncertain. Finally, the scales used for both the extinction coefficient and the extinction coefficient ratio are different and differently related for all cases, making the establishment of parallel behaviour uncertain. Hence, drawing any conclusion on these values, taking into account the high uncertainty on the value of the maximum extinction [ratio] (See comment on L. 160, p.5) looks particularly hazardous.

*The first data is the first data shown in Figure 7. As discussed in section 2, the temporal sampling from both SAGE II and III is sparse particularly in the tropics and this is one of several challenges in using this data in this analysis. Within these data sets, we observe that aerosol extinction coefficient levels at a given altitude and latitude slowly vary with time even apart from recent volcanic activity due to the recovery from past volcanic activity and seasonal processes. For the events discussed here, due to the timing of the events, these changes are very small compared to the volcanic events that follow and, in terms of the calculation of perturbation values, the exact background level has only a secondary effect on the*

*calculated values. As a result, the timing of the 'before' samples does not materially affect these results. We have clarified this in the text. The scales used in the plots are selected to highlight the behavior of each event and is necessary considering the several order-of-magnitude differences between events. The Ruang and Manam events occur during the 50% duty cycle portion of the SAGE II record and thus we have less data during this period. The base values are the closest in time available and occur during a volcanically quiet period and are adequate for this analysis. We have changed the reference from 'baseline' to 'before'*

L. 184-186, p. 6: See previous remark.

*See previous reply*

L. 197, p. 6-L. 201, p. 7: Beyond the uncertainties related to the values plotted in Figure 8b (See comment on L. 179-181, p. 6), this model is so coarse, ignoring the fact that some eruption produce more sulfur gases and some other ones more ashes, ignoring all effects related to the geolocation and consequent dynamical features of the plume dispersion as well as seasonal effects, and ignoring intrumental limitations (no- tably SAGE II's "blindness" in the early phase of the Pinatubo eruption), that it seems absolutely premature to draw any conclusions from these 10 cases among which only a handful of the considered ones follow a nice flat curve. Applying such relationship to other data sets from limb sounders where retrieved extinction values depend yet on a set of assumption, is at risk to get quite far removed from the truth. Instead of suggesting using this tool for the evaluation of interactive aerosol modules for GCMs and ESMs, it would be useful, on the contrary, to use GCMs with elaborate aerosol microphysical models to assess if this relationship is sufficiently grounded to be used elsewhere.

*As we discuss in section 2, with few exceptions, the ability to infer the detailed processes within a volcanic*

*plume and in stratospheric aerosol from SAGE-like measurements are limited. However, the instruments do not care about the processes that create the optical properties that they measure. The measurements themselves are robust and the features we show in Figure 8 are readily apparent in the most cursory examination of the data. It would be convenient to have many more events on which to examine this relationship but such data are not available. Still, the observed relationship occurs in 80% of the data available which we find compelling. We have modified the text to reflect that, in Figure 8b, perturbation ratio is reasonably well sorted in perturbation extinction and removed the reference to linear or curved lines as probably being a little too enthusiastic. We did not originally plot a line or provide a fit in this case in part to reflect the low data amount and the uncertainty in the individual points. We agree that understanding is most compelling when there is closure between observations and models. That is why we suggest that these data provide an opportunity for closure tests. We have modified the final sentence in this paragraph to clarify our intent.*

L. 206-207, p. 7: If two closely related eruptions from a same volcano, occurring within such a short period of time, have yet to be splitted in a "regular case" and an "outlier", it becomes very hard to find any foundation in the proposed relationship. Even if other observations could bring some evidence that the plume composition, due to this quick succession, was very different in both cases, such argument could not be invoked since this kind of consideration has been excluded in this study.

*In this context, this comment reflects that the second event may, at first glance, be an outlier. Further analysis shows that it is not. Generally, there is no reason to believe that eruptions from a volcano will necessarily be the same when separated months or years.*

L. 210-213, p. 7: Choosing the "initial state" of the second eruption of Ambae (with a value of 4.9 for

the extinction ratio) is a highly speculative exercise ignoring the fact that the microphysical evolution is highly perturbed by the addition of fresh material. In no way, the situation corresponding to this value of 4.9 for the extinction ratio can be considered as a "baseline". Furthermore, the given values are quite arbitrary: the value of 5 is never reached by the extinction ratio (the maximum is about 4.7), and the extinction ratio decreases far below 4.1, toward a value lower than 3.5 and that cannot be estimated from the limited curve shown in Figure 7. Hence, putting both Ambae eruptions again (and in contradiction to what is observed in L. 206-207, p. 7, see comment) in the category of "regular cases" is particularly dubious.

*Again microphysical processes that may or may not be occurring do not change the observations. The April eruption of Ambae is overwhelmed by the much larger second event and we can get essentially the same perturbation values whether we use the minimum just prior to the second eruption or the values prior to the initial Ambae eruption as the 'before' values. We have clarified that the values reflect the initial changes observed following the eruptions. In retrospect, the 4.9 value appears to occur after the July Ambae eruption and we have modified the text to reflect that the extinction ratio changes from around 4.5 just prior to the second eruption, to 4.9 with the earliest observations of the new aerosol and then to 4.1 when the aerosol extinction coefficient is a maximum. We have changed the initial extinction coefficient ratio from 'nearly 5' to 4.7.*

L. 220-221, p. 7: This looks like a fast conclusion about a situation where microphysics, dynamics, and multiple injections of volcanic matter combine in a complex way, as illustrated in Figure 9.

*The first Ambae eruption is inconsequential to this depiction. While we are aware that there are complex processes going on, this is what is observed by the instruments and it is consistent with what is depicted in Figure 8.*

L. 223, p.7: "the main aerosol layer", "all parts of the volcanic cloud": please clarify (See also comment on L. 162, p.6).

*Updated to 'the densest part of the volcanic plume'*

L. 226-228, p. 7: What do the authors mean by "the first observations"? Also, the end value is rather 2.9 than 3.

*Added 'of the plume'*

L. 228-229, p.7: Where do the authors find this value of 2.7? Such value is never reached in Figure 7, nor is indicated in Figure 8. Is the "perturbation extinction ratio" another quantity than the ones illustrated in these figures?

*We've updated the value to 2.9 and the later value from 4.3 to 3.9.*

L. 230, p. 8: What is a "compact extinction coefficient [ratio]"?

*Updated to indicate that the scatter of the data is mostly compact.*

L. 230-235, p. 8: It is surprising to give such an importance to outliers that in other circumnstances would be fastly overlooked.

*There are sufficient numbers of observations showing enhanced extinction and low extinction ratio to impact the averages (unlike those in Figure 5 for Ruiz) so their presence cannot be ignored.*

L. 239, p. 8: What do the authors mean by "the spread of the Nevado del Ruiz, Cerro Hudson and Raikoke eruptions"?

*Changed to 'differences between'*

L. 243-251, p. 8: This part of the text looks like a suite of speculations without attempt to analyse them seriously, making this enumeration not very useful.

*We feel obligated to enumerate issues that may impact the results we show but cannot currently explain.*

L. 257-259, p. 8: The sentence is unclear. Please rephrase.

*Rewritten*

L. 263-266, p. 9: The sentence is unclear. Please rephrase.

*Rewritten*

L. 286, p. 9: What do the authors mean by "identical perturbations"?

*This has been clarified in the text.*

L. 285-293, p. 9 and Figure 11: Basically, the extinction coefficient at 525 nm and 1020 nm and hence they ratio, can be exactly calculated by Mie theory for fixed values of the aerosol volume density and for single-radius particles. Hence, the differences found in Figure 11 for Nevado del Ruiz and Ambae only shows in which extend this simpel
model of single-radius particles departs from the reality. Furthermore, the authors mention coagulation effects, but ignore sedimentation which is nevertheless crucial after a large eruption, and high from the early post-eruption period.

*We did not make the purpose of this simple model sufficiently clear and thus have expanded the discussion of its rationale. We do not intend for it to capture all aspects of microphysical processes going on following an eruption (it is indeed far too simple for that). We do use it show to*

*demonstrate why we believe that the observations are consistent with the idea of the nucleation of many small particles following a small to moderate eruption (that initially may be invisible to SAGE-like measurements) followed by coagulation to larger, optically significant particles. There is an extensive discussion of the limitations of this model in the third paragraph of this section which highlight it shortcomings.*

*We had extensive discussions among the authors regarding what the observations mean and the nucleation of many small particles was the only process that we can see capable of producing what is measured by the instruments. We have highlighted the lack of sedimentation as a shortcoming particularly for large eruptions though our focus remains on small to moderate eruptions. We believe that it is reasonable to offer an explanation of what we believe the measurements mean.*

L. 289, p. 9: There is no single timescale used in Figure 11, nor in Figure 2, and the authors should thus not use this concept alone.

*There is no timescale for Figure 11 or 2. Using particle size as a pseudo-time scale in Figure 11 is explained in the text and shortcomings related to this discussed. This is not relevant to figure 2 which is simply a plot of Mie extinction kernels as a function of radius.*

L. 305-307, p. 10: From the simplicity of this model and the fact that important aspects are neglected (e.g. the absence of sedimentation which critically influences the extinction and extinction ratio), I do not think that any conclusion can be drawn about inferring primary microphysical effects from these SAGE II and III/ISS observations.

*We think we have adequately demonstrated why we believe our interpretation of the results is reasonable.*

*However, it is clear that it is only through closure between observations and modelling that confidence in this inference can be obtained.*

L. 323, p. 10: Which "measurement paradigms" are the authors talking about?

*Changed to simply 'observations.'*

L. 327-328, p. 11: Using a tool to assess the data quality of a data set is only mean- ingful if this tool is not based on coarser assumptions and approximations as the ones leading to this data set. I am not sure that this is the case in the present work.

*We do not understand what the reviewer intends here. In this case, we are referring to using outcomes from this study as an aid to improving data quality for instruments like OSIRIS.*

L. 331-333, p.11: Among the different applications proposed by the authors, the use of an extinction ratio value inferred from a relation established from the points 6-5-8- 7-0-2 in Figure 8b to fix the size parameter used in the OSIRIS retrieval (e.g. through the Angstrom coefficient) seems to be the only one for which this simple model could have a real added value, in my opinion. It is really a pity if this model even cannot be beneficial in this framework.

*The use of these results in OSIRIS retrievals is an on-going study which we hope will result in positive improvements in the OSIRIS aerosol data products in the future. WE have indicated this in the text.*

Technical corrections

Caption Figure 9: It would be useful to indicate here the latitude of the Ambae volcano.

*Done*

L. 216-217, p. 7 and caption Figure 9: the indications of wavelength and time are not consistent between the text and the caption. Please specify the exact time duration (start and end time).

*Corrected to September.*

---

## Author Comment (AC3) · 15 Oct 2020

General comments

The paper describes the relationship between the perturbation in the aerosol extinction  coefficient and the extinction ratio or particle size in the early months following small  to midsize volcanic eruptions. The authors acknowledge the limitation of the analysis,  which were restricted to early

months following volcanic eruptions while tracking the main layer, mainly because of the presence of significant amount of ash.  I believe that the analysis presented here are incomplete and the paper would benefit from expanding the analysis to a longer period (not only the early months) and wider range of altitudes, rather than the peak of the aerosol layer. In addition, the paper needs clear and defined objectives (see my comment below).

Specific comments

Page 3, L64-66: 'space-based missions are mostly limited to single wavelength mea- surements associated with instruments such as . . .' The statement is only valid for the datasets used in GloSSAC climatology. CALIPO routinely provides aerosol measure- ments at two wavelength 532 and 1064 nm, while OSIRIS can provide measurements at 750 and 1530 nm, which were used to derive stratospheric aerosol particle size in- formation (Rieger et al., 2014). In addition, MAESTRO (McElroy et al., 2007), GOMOS (Vanhellemont, et al., 2016), and SCIAMACHY (Taha et al., 2010; Malinina et al., 2019) can also provide aerosol measurements at multiple wavelengths. While the quality of the measurements is debatable, its existence is not.

*We've included some additional sources of stratospheric aerosol measurements and clarified the data available.*

Page 3, L73-74: 'It is also, until the start of the SAGE III mission, a period where the long-term stratospheric record is less robust due to the lack of global multiwavelength measurements of aerosol extinction coefficient.' Again, the Authors are describing the GloSSAC dataset rather than the global stratospheric records. See previous point.

*This point is similarly clarified.*

Page 3, L76-81: 'Thus, the original aim of this work was to understand how volcanic events manifest themselves in SAGE II/III observations with the goal of 1) inferring the uncertainty in single wavelength space-based data sets that use a fixed aerosol size distribution as a part of their retrieval algorithm such as the OSIRIS and CALIOP and 2) infer how well the wavelength dependence can be estimated for these single wavelength measurements.' The authors failed to address both objectives and I don't see how the paper's findings, in its current form, can be of any use to these instruments because of the limited analysis shown her. I suggest either expanding the scope of the work to address those objectives or revising it to more realistic objectives.

*We've revised the goals of the paper (and moved the discussion of them primarily to the introduction) as the need to characterize the way in which small-to-moderate volcanic eruptions manifest themselves in SAGE-like observations and that the goal for applications to OSIRIS and other data sets is a longer term objective.*

Table 1: 'Volcanic eruptions and smoke events that significantly impact stratospheric aerosol levels in the Version 2.0 of the GloSSAC data set' Figure 1 shows very low aerosol and no impact of any of the volcanoes listed between 1998-2004, which implies that none of these eruptions reached the stratosphere. I suggest revising Table 1 by removing all volcanic eruptions that is not seen by GloSSAC. In addition, add the eruption altitude, similar to Table 2.

*Since the Table refers only to Figure 1, the altitude of the event other than being stratospheric isn't relevant.*

Figure 3: Unlike the rest of the analysis shown in this paper, the figure is for GloS- SAC zonal mean aerosol stratospheric optical depth and ratio rather than extinction coefficient and

ratio for the peak aerosol layer. I suspect that the extinction ratio is inaccurate, given that SAGE II measurements were missing following the early months of the eruption, and the dataset was mostly reliant on single wavelength Lidar mea- surements. For the sake of consistency, I suggest using SAGE II measurements of the aerosol extinction and extinction ratio, similar to Figures 4-7.

*GloSSAC data is used for a relatively minor and a broadly accepted idea that the wavelength dependence of aerosol extinction/optical depth becomes very small (near 1) very quickly following the Mt. Pinatubo eruption. The data the reviewer requests is partially in Figure 7b. We really don't see any reason to change this figure. The process for filling missing data in GloSSAC is described in (Thomason et al., 2018;Kovilakam et al., 2020).*

Figure 4a: Can you use different color for the extinction values at 20.5 km?

*All data is at 20.5 km.*

Figure 4b: Can you plot all data shown in Figure 4a while using different color for extinction ratio at 20.5 km?

*All data is at 20.5 km.*

Figures 4, 6, and 7: The Days label is confusing. Can you use month/year or something similar?

*We have explained the coordinate system in the caption more fully.*

Figure 5: change 'Same data as shown in Figure 3' to Same data as shown in Figure 4a or 4b. In addition, can you specify if the data shown are for 20.5 km or all altitudes? If so, can you use different color for 20.5 km.

*Done*

Page 6, L189 'the maximum extinction coefficient at 525 nm does not necessarily occur at the same altitude or time as the maximum in 1020 nm extinction coefficient'. This is all the more reason to track the plume at different altitudes/zones rather than the maximum extinction value.

*See response below.*

Figure 7: I suggest over plotting points symbol to show the number of points used in each figure, adding a second vertical line for Ambae and Ulawun denoting the second eruption date, and using the same x-axis scale for all figures.

*Done.*

Figure 9 and page 7, L220 'It is clear here that the maximum in the extinction ratio lies below the main peak in extinction coefficient in the tropics and, notably stretches to higher southern latitudes and the maximum values actually occurs near 30° S de- spite more inhomogeneous conditions at this latitude than in the tropics' and page 8 L240: 'Both eruptions show increased aerosol extinction coefficient ratios away from the main aerosol peak suggesting, at least in part, behavior more consistent with most eruptions.' This is interesting observation that raises more concerns about the analysis shown in Figures 6 and 7. Is it possible that tracking the maximum extinction value is not the best approach as it might bias the outcome, especially where the aerosol extinc- tion is very large? Very large extinction values can be caused by the presence of Ash particles, or it is an artifact in SAGE measurement when the volcanic plume is localized and spatially inhomogeneous. In addition, the result results shown here can be easily biased by

SAGE limited coverage. Perhaps repeating Figure 7 using zonal means at different altitudes and extending period of the analysis can produce a more consistent relationship between the aerosol extinction and extinction ratio perturbations.

*We agree that tracking the maximum in extinction raises the likelihood that the observations include ash particles. However, material is transported out of the latitude band where the event occurred, it is about as likely to include ash as sulfuric acid aerosol. In addition, as the observations become further from the point of injection, there is a greater likelihood that a SAGE II profile with a volcanic signature at a particular altitude consists of an inhomogeneous mix of air unaffected by the eruption with the volcanic plume through either mixing or the plume only occupying a part of the measurement volume. This makes the interpretation of the an extinction measurement pair more problematic much in the way that SAGE observations of water clouds are better interpreted as aerosol/cloud mixed extinction coefficient values rather than purely 'cloud' extinction coefficient (Thomason and Vernier, 2013). While this is always a problem, we have focused on the densest part of the plume as a means to mitigate this effect. We have included mention of this issue in the paragraph originally beginning at L81 (It should be clear…).*

Page 8, L246: 'peak extinction level as essentially all the data follows the mean rela- tionship in Figure 7g.' The sentence is unclear. This correct for Ulawn, however, the paragraph is discussing Raikoke eruption. Please revise the sentence.

*We have clarified this statement.*

Page 8, lin3 253: 'It is also possible that a pyrocumulus event, that occurred in Alberta, Canada just prior to the Raikoke eruption, plays a role in the evolution of extinction fol- lowing this event.'. In addition to Alberta, there was a second PyroCb event in Siberia, Russia in July 2019, that reached the stratosphere and was also seen by SAGE III/ISS (https://directory.eoportal.org/web/eoportal/satellite-missions/i/iss-sage-3, Figure 7). It is more likely that the smoke aerosol interfered with the Raikoke analysis. If the two different aerosol layers were separated (which is most likely), then repeating the analy- sis at different altitudes instead of tracking the peak extinction can explain the behavior seen in Figure 7h.

*We have found the separation of the pyrocumulus events and the volcanic events in the lower stratosphere to be particularly complicated. The sulfuric acid aerosol layer from the Raikoke eruption was impacted by smoke injected from the Canadian and Siberian wildfires. The degree to which the two mixed is uncertain, but the data clearly show some mixing. Despite a secondary cloud (most likely smoke) breaking from the main Raikoke layer and moving south, our research indicates that smoke likely remained within the main Raikoke peak until it was fully dispersed. Therefore, repeating the analysis does not deconvolve these two components. We have a manuscript in preparation dealing with some of the unusual aspects of this eruption which is a really interesting event or set of events given the smoke plumes.*

Section 4: The first paragraph leading to the aerosol perturbation model relies on unsupported speculations (by the authors own admission), and it can benefit from the addition of few references that support those assumptions.

*References added.*

I also find the whole discussion regarding the model calculation and figure 11 confusing and I can't

relate the figure to what was presented in the previous section. The aerosol extinction ratio perturbations for Ambae are almost double those for del Ruiz eruption and figure 11b in particular shows large extinction ratios (>5) not seen by any of the cases shown in this paper. Close inspection of both figures indicate that the extinction ratio perturbation is very sensitive to the baseline ratio, and if Manam or Ulawn baseline values were used, the extinction ratio values would've been even higher.

*That the differences for the same perturbations manifest themselves differently depending on the pre-eruption background is one of the points behind the model and is consistent with what is shown in Figure 8. The other key reason for including is to provide a rationale for our interpretation of observations as suggesting the homogeneous nucleation of many small particles followed by coagulation. The increase in extinction ratio as the aerosol perturbation becomes relevant followed by a relaxation to smaller values. We have substantially rewritten the introduction to this section to explain the goals and rationale of this section. Some extinction ratio values exceed those observed herein are associated with hypothetic 'volcanic' events substantially greater those observed in the events in Figure 11.*
* * *
Kovilakam, M., Thomason, L., Ernest, N., Rieger, L., Bourassa, A., and Millán, L.: The Global Space-based Stratospheric Aerosol Climatology v2.0, Earth System Science Data Discussions,

10.5194/essd-2020-56, in review 2020.

Thomason, L. W., Ernest, N., Millán, L., Rieger, L., Bourassa, A., Vernier, J.-P., Manney, G., Luo, B., Arfeuille, F., and Peter, T.: A global space-based stratospheric aerosol climatology: 1979-2016, Earth System Science Data, 10, 469-492, 2018.

---

## Referee Report (RR1)

I recommend publishing the manuscript subject to the following technical corrections:

L45: change 'mall-to-moderate' to small-to-moderate

L90: add OMPS LP (Loughman et al, 2018 or Taha et al., 2020)

L173: 'and are re usually less than' remove re

L213: 'within a4-km' add space after a

L251: 'volcanic eventsin' add space after events

L332: 'It is also possible that pyrocumulus events in Alberta, Canada and Siberia occuring around the time of the Raikoke eruption (Yu et al., 2019), play a role in the evolution of extinction following this event'

I don't see Yu et in the reference list. Perhaps Kloss et al., 2020 is more appropriate.

Refrences:

Kloss, C., Berthet, G., Sellitto, P., Ploeger, F., Taha, G., Tidiga, M., Eremenko, M., Bossolasco, A., Jégou, F., Renard, J.-B., and Legras, B.: Stratospheric aerosol layer perturbation caused by the 2019 Raikoke and Ulawun eruptions and climate impact, Atmos. Chem. Phys. Discuss., https://doi.org/10.5194/acp-2020-701, in review, 2020.

Loughman, R., Bhartia, P. K., Chen, Z., Xu, P., Nyaku, E., and Taha, G.: The Ozone Mapping and Profiler Suite (OMPS) Limb Profiler (LP) Version 1 aerosol extinction retrieval algorithm: theoretical basis, Atmos. Meas. Tech., 11, 2633-2651, doi:10.5194/amt-11-2633-2018, 2018.

Taha, G., Loughman, R., Zhu, T., Thomason, L., Kar, J., Rieger, L., and Bourassa, A.: OMPS LP Version 2.0 Multi-wavelength Aerosol Extinction Coefficient Retrieval Algorithm, Atmos. Meas. Tech. Discuss., https://doi.org/10.5194/amt-2020-329, in review, 2020.

---

## Author Response (AR2)

**Response to Reviewer 1**

Minor Comments:

1) All new text (underlined) should be re-read because I have found many typos. Please find some in the following but the list is not exhaustive

*A few further typos have been corrected .In addition, an update for the Kovilakam reference is included.*

2) Abstract, L1-5: please break up this very long sentence into two shorter sentences.

*Done*

3) Abstract: "They may also represent be a distinct avenue...", there is a "be" to be suppressed?

*Yes, removed.*

4) Introduction, Page1: "mall-to-moderate" -->"small-to-moderate"

*Done*

5) "OSIRIS, 2002- present)" --> "OSIRIS (2002- present)

*Done*

6) Introduction, Page 2: "... and apparent aerosol particle size...", please define "apparent" (e.g.: "using the spectral variability of the aerosol extinction as a proxy" or similar)

*Text is clarified as the reviewer suggested.*

7) Results, Page 4: "The spread early in extinction coefficient and in extinction coefficient ratio is primarily due to inhomogeneity in the volcanic aerosol within the analysis area.", it has been recently shown how spatial dispersion and temporal evolution (i.e. the action of in-plume physico-chemical processes) "smooth" volcanic plumes (https://www.nature.com/articles/s41598-020-71635-1). That's a well-known behaviour for volcanologists but not as well clear in atmospheric science works, so I suggest to mention this using the above reference.

*This reference is now included in the text.*

8) Page 5: "In any case, these points are rare and only occur in the first month following the eruption.", why not mentioning that this can be due to removal of larger particles from the plume (sedimentation)?

*Text clarified as the reviewer suggests.*

9) "...a4-km..." --> "a 4-km"

*Fixed*

10) "...(solid) and extinction coefficient ratio (dash) are well...", I would not mention line style in the text (this is something that is more for the figure caption)

*Descriptors are removed.*

11) "We will discuss some of these events in more detail below.", It is said here that individual events are discussed in the following. Nevertheless, before the specific discussions of the events, there still is a full paragraph of general discussions. I'd suppress this sentence here and add something like "We now discuss individual some events" when you start this part (i.e. for Ambae)

*Wording changed as suggested.*

12) Equation at the end of Page 5: why this and the following equations are not numbered? Also, "k(max)", is this what has been called "after" (and not "max") above in the text? Please be consistent.

*Equation numbers added and updated further along in the paper...*

13) Page 6: ""eventsin" --> "events in"

*Corrected.*

14) Equation for "ratio": I would use a more specific name for this parameter, to avoid confusion between the ratio of the perturbations and the ratio of the extinctions themselves (also called ER)

*Changed to 'perturbation ratio.'*

15) Page 7: "For instance, for Raikoke, we cannot exclude the possibility of the presence of small amounts of ash embedded in the main aerosol layer with the sulfuric acid aerosol influencing the extinction coefficient and ratio. The presence of ash following the Raikoke eruption has been inferred above 15 km and perhaps as high as 20 km ...", this preprint (ACPD) on Raikoke eruption observed with SAGE (among other data) is now available: https://acp.copernicus.org/preprints/acp-2020-701/ . This work also suggests possible presence of ash, and more absorbing material, in the Raikoke plume, that can be used to confirm your hypothesis

*Reference added.*

16) Page 8: "However, as they coagulate into steadily larger particles (possibly also consuming small-sized aerosol present in the pre-existing aerosol layer)...", I would not be so categorical here on the fact that the microphysical processes that modify these size distributions are just coagulation; I'd mention also condensation processes here.

*Text clarified as suggested.*

17) Figure 7: "Which is the starting date for each abscissa? Please double check the position of the red dashed lines: e.g. for Pinatubo, the AE perturbation seems to anticipate the eruption."

*A more complete explanation of the 'day' definition is now included in the caption.*

**Response to Reviewer 3**

I recommend publishing the manuscript subject to the following technical corrections:

L45: change 'mall-to-moderate' to small-to-moderate

*Done*

L90: add OMPS LP (Loughman et al, 2018 or Taha et al., 2020) L173: 'and are re usually less than' remove re

*Done*

L213: 'within a4-km' add space after a

*Done*

L251: 'volcanic eventsin' add space after events

*Done*

L332: 'It is also possible that pyrocumulus events in Alberta, Canada and Siberia occuring around the time of the Raikoke eruption (Yu et al., 2019), play a role in the evolution of extinction following this event'

I don't see Yu et in the reference list. Perhaps Kloss et al., 2020 is more appropriate.

*Yu reference is now included in the reference list.*

[revised manuscript text omitted]
 compare to the low frequency during particular the last decade of the SAGE II mission, it is clear that there is a significant qualitative difference in the stratospheric aerosol variability in between the two periods. After the end of the SAGE II mission in 2005 and until the start of the SAGE III mission, long-term stratospheric record is less robust partly due to the limited global multiwavelength measurements of aerosol extinction coefficient.

It should be clear from the outset that the solar occultation measurement strategy is, in general, not conducive to process studies and understanding the distribution of aerosol following highly localized events like volcanic eruptions. Following these sorts of events, we observe that SAGE observations have a high zonal variance in the data compared to more benign periods where the zonal variance is often not much larger than the measurement uncertainty particularly in the tropics (Thomason et al., 2010). The events we discuss below are not sampled in a temporally uniform way and the time between an eruption and the first SAGE II observations at the relevant latitudes varies from a few days to more than a month.  This is an outcome of the sparse spatial sampling characteristic of solar occultation with latitudinal coverage dictation by orbital and seasonal considerations and a given latitude is measured at best once or twice per month. In addition, with 15 profiles per day with 24 degrees of longitude spacing, the sampling is sparse in longitude even when latitudes of interest are available.  Furthermore, aerosol properties in a single profile at a single altitude are the average of multiple samples along different line-of-sight paths through the atmosphere such that the spatial extent of a measurement at an altitude extends over hundreds if not thousands of square kilometers (Thomason et al., 2003).  This large measurement volume increases the possibility that only part of a SAGE II observation's measurement volume that will actually consist of volcanically-derived material. This makes the interpretation of the an extinction measurement pair more problematic much in 
[revised manuscript text omitted]

---

## Author Response (AR3)

Each of the technical corrections below have been incorporated into the manuscript and underlined in this version. I also made a wording change starting at line 81 for clarity as well.

P1, L23: such the -> such as the

P3, L99-101: Sentence not clear. Please check and correct.

P3, L103: add "the" before "long-term"

P3, L117-119: Sentence not clear. Please check and correct.

P3, L124: measurements are high -> measurement are of high

P4, L180: in used -> is used

P5, L200: Either plural or singular should be used here. Thus, either " a very high aerosol extinction coefficient" or "very extinction coefficients".

P5, L200: extinction coefficient ratio or ratios. This depends on what you decide on for the part before.

P6, L270: ratio appears twice, thus one is obsolete.

P6, L292: Parentheses around "Figure 8b" are missing.

P7, L320: behaviour -> a behaviour

P8, L353: times scales -> time scales

P9, L423: condensing homogeneously -> nucleating homogeneously ?

P9, L428: Plural better here? Thus, "coefficients"?

Further, I would like to ask you to use a larger font and the typical manuscript style double space lines for your next manuscripts. The manuscript in the present form was really hard to read with this incredibly small font and tight space between lines.

[revised manuscript text omitted]